# Active Reasoning in an Open-World Environment

**Manjie Xu** [1,†]
manjietsu@bit.edu.cn

**Guangyuan Jiang** [2]
jgy@stu.pku.edu.cn

**Wei Liang** [1,3,✉]
liangwei@bit.edu.cn

**Chi Zhang** [4,✉]
zhangchi@bigai.ai

**Yixin Zhu** [2,✉]
yixin.zhu@pku.edu.cn

[1] School of Computer Science & Technology, Beijing Institute of Technology
[2] Institute for AI, Peking University
[3] Yangtze Delta Region Academy of Beijing Institute of Technology, Jiaxing, China
[4] National Key Laboratory of General Artificial Intelligence, BIGAI

**https://sites.google.com/view/conan-active-reasoning**

## Abstract

Recent advances in vision-language learning have achieved notable success on *complete-information* question-answering datasets through the integration of extensive world knowledge. Yet, most models operate *passively*, responding to questions based on pre-stored knowledge. In stark contrast, humans possess the ability to *actively* explore, accumulate, and reason using both newfound and existing information to tackle *incomplete-information* questions. In response to this gap, we introduce 🔍 **Conan**, an interactive open-world environment devised for the assessment of *active reasoning*. 🔍 **Conan** facilitates active exploration and promotes multi-round abductive inference, reminiscent of rich, open-world settings like Minecraft. Diverging from previous works that lean primarily on single-round deduction via instruction following, 🔍 **Conan** compels agents to actively interact with their surroundings, amalgamating new evidence with prior knowledge to elucidate events from incomplete observations. Our analysis on 🔍 **Conan** underscores the shortcomings of contemporary state-of-the-art models in active exploration and understanding complex scenarios. Additionally, we explore *Abduction from Deduction*, where agents harness Bayesian rules to recast the challenge of abduction as a deductive process. Through 🔍 **Conan**, we aim to galvanize advancements in active reasoning and set the stage for the next generation of AI agents adept at dynamically engaging in environments.

## 1 Introduction

Active interaction with the environment is fundamental to human understanding of the world around us. Both neural and behavioral studies indicate that through active engagement with their surroundings, humans garner critical insights and foster a profound understanding of complex phenomena (Goodale and Milner, 1992; Rizzolatti et al., 1997; Rieber, 1996). When confronted with partial or ambiguous data, our innate response is to seek supplementary evidence, hypothesize, and put forth possible explanations, sometimes even reevaluating initial assumptions (Yuan et al., 2022). This iterative process persists until a satisfactory resolution emerges.

The process of formulating theories based on observations and prior knowledge is classically termed as abductive reasoning or simply, abduction (Peirce, 1965; Douven, 2021). A topic of enduring

---

[†] Work done while M. Xu was an intern at Peking University.

37th Conference on Neural Information Processing Systems (NeurIPS 2023).

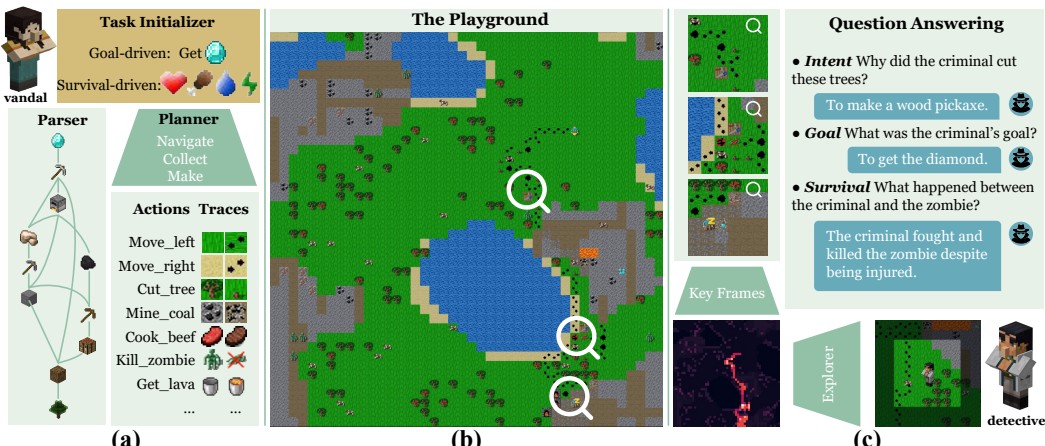

Figure 1: **An example of 🔍 Conan, an open-world environment for active reasoning.** (a) 🔍 Conan initialization. A *vandal* is randomly assigned a task from the task space while keeping alive. A probabilistic parser, utilizing a knowledge graph, selects a sequence of subgoals to fulfill the main objective. This decision is then conveyed to a planner which, in turn, invokes heuristic policies to execute atomic actions. Some of these actions leave discernible traces within the environment. (b) 🔍 Conan playground with traces. (c) 🔍 Conan questions. Here, a *detective* is spawned and is tasked with answering queries. It does so by actively exploring the environment, connecting keyframes, and reaching conclusions.

interest among psychologists, abduction is perceived as a cornerstone of human cognitive processes. Historical and contemporary studies have delved into its cognitive mechanisms (Josephson and Josephson, 1996; Thagard, 1988; Peirce, 1965), practical applications (Hobbs et al., 1993; Shank, 1998), and ties to scientific thinking and decision-making (Hanson, 1965; Gigerenzer and Gaissmaier, 2011; Zhang et al., 2021a). With growing momentum in the machine learning sphere, recent years have witnessed the advent of dedicated benchmarks and models accentuating abductive reasoning (Bhagavatula et al., 2019; Kayser et al., 2021; Hessel et al., 2022; Liang et al., 2022).

However, the bulk of prior work in this domain relies heavily on a single-round, *passive* question-answering paradigm that offers complete information. This setup often sees an agent simply responding to queries, leveraging vast pre-trained knowledge, as evidenced by the latest strides in language and vision-language learning. Recent progress in the field has notably already improved performance in such complete-information information question-answering. Contrarily, humans demonstrate a far more nuanced approach when navigating abductive scenarios with incomplete data (Edmonds et al., 2018). We *actively* engage, explore, gather, and reason, drawing from both new information and prior knowledge. Our iterative approach allows for continuous refinement based on newly acquired evidence (Oaksford and Chater, 1994; Bramley et al., 2017; Edmonds et al., 2019, 2020).

To capture the dynamic and exploratory essence of abductive reasoning—termed herein as *active reasoning*—we present 🔍 Conan, a new open-world environment tailored for abductive reasoning. Standing head and shoulders above traditional single-round passive reasoning benchmarks, 🔍 Conan boasts an open-world arena, urging agents to actively probe surroundings and engage in multi-round abductive inferences, all while leveraging *in-situ* collected evidence alongside pre-existing knowledge.

At its core, 🔍 Conan is conceived as a detective game, transmuted into a question-answering challenge. Here, the *detective* is tasked with a query and an "incident scene" riddled with traces left by a *vandal*. Given the initial paucity of conclusive information, the *detective* must embark on an in-depth exploration of the scene. As the inquiry progresses, the *detective* has the opportunity to actively scout its environment, continually reshaping and honing its hypotheses, especially when new revelations potentially contradict the prior hypothesis. Furthermore, we meticulously craft questions within 🔍 Conan to span various levels of abstraction, from localized intentions (Intent) to overarching objectives (Goal) and survival states (Survival).

To probe the proficiency of active reasoning, we evaluate state-of-the-art Reinforcement Learning (RL) and multimodal question-answering models on 🔍 Conan. Our observations highlight an intriguing dichotomy: while these cutting-edge models exhibit prowess in addressing low-level, short-term tasks, they struggle with multi-round environmental interactions and high-level abductive reasoning.

A plausible root of this challenge could be the absence of structurally represented knowledge. Predicated predominantly on associative training, these agents are versed in correlating traces with responses without genuinely internalizing holistic world models. In sharp contrast, humans seamlessly navigate abductive reasoning by forecasting potential trajectories leading to a perceived outcome. This intricate dance gradually transmutes from abductive to deductive reasoning, where humans harness their innate understanding of causality to deduce and mirror observed patterns. In our pursuit to mirror this quintessential human trait, we integrate Abduction from Deduction (AfD) into 🔍 **Conan** via a Bayesian approach. Experimental results underscore the efficacy of AfD, indicating a substantial avenue for bolstering agent adeptness in 🔍 **Conan**.

To sum up, our work makes the following three contributions:

- We usher in the novel domain of *active reasoning*, underscoring the indispensable roles of active exploration and iterative inference in abductive reasoning. This paradigm shift transforms traditional single-round passive question-answering paradigms into a more immersive format, compelling agents to actively engage with the environment to procure pivotal evidence.
- We introduce 🔍 **Conan**, a new environment tailored to evaluate the abductive reasoning ability of current machine learning models within dynamic settings. 🔍 **Conan** surpasses its predecessors that hinge on step-by-step deductive reasoning, revealing the limitations of present-day models.
- We formulate a new learning method for abduction, AfD, grounded in Bayesian principles. This framework elegantly reformulates abduction into deduction, proving instrumental in navigating the complex active reasoning challenges posed by 🔍 **Conan**.

## 2 Related Work

**Machine Abductive Reasoning**  Abductive reasoning, foundational to human cognition, is crucial for scientific exploration, decision-making, and problem-solving (Peirce, 1965; Magnani, 2011). In the Artificial Intelligence (AI) landscape, there is a rich history of efforts to equip machines with this ability, where they use prior knowledge and sparse observations to hypothesize amidst uncertainty (Josephson and Josephson, 1996; Xu et al., 2023). Key developments span logic-based abduction (Kakas et al., 1992; Poole, 1993) and hybrid neural-symbolic methods (Rocktäschel and Riedel, 2017; Zhang et al., 2021b; Li et al., 2022, 2023). With computational progress, Large Language Models (LLMs) have effectively addressed several challenges through text generation, exhibiting outstanding performance (Brown et al., 2020; OpenAI, 2023; Thoppilan et al., 2022). Modern research usually frames abductive reasoning within natural language understanding (Bhagavatula et al., 2019) or multimodal vision-language integration (Hessel et al., 2022; Liang et al., 2022). However, there is still a notable gap: many benchmarks lean heavily on deduction, sidelining abduction's interactive essence. Our work addresses this gap, emphasizing the core of *active reasoning* in abductive contexts.

**Embodied Question Answering**  Embodied question answering enhances traditional Visual Question Answering (VQA) by placing agents in interactive environments (Johnson et al., 2017; Das et al., 2018; Gordon et al., 2018; Yu et al., 2019). In 🔍 **Conan**, agents actively explore to gather data, preparing them to solve abductive questions based on partial information. Unlike standard embodied question-answering frameworks (Das et al., 2018; Gordon et al., 2018; Yu et al., 2019), where questions become simple instructions for agents, 🔍 **Conan** introduces complexity: (i) its questions, rooted in high-level intent and goals, resist simple decomposition into a series of actions; (ii) agents in 🔍 **Conan** act as detectives, constantly hypothesizing from observations and prior knowledge, and iterating their strategies in light of new data. For a comprehensive comparison of 🔍 **Conan** with other benchmarks, see Tab. 1.

## 3 The 🔍 Conan Environment

🔍 **Conan** is crafted as an interactive question-answering environment aimed at evaluating a machine's active abductive reasoning capacity, as depicted in Fig. 1. Building on the foundation of the Crafter (Hafner, 2021), 🔍 **Conan** evolves into a detective game featuring two agents: the *vandal* and the *detective*. The gameplay kickstarts with the *vandal* undertaking a randomly designated task, leaving behind traces for the *detective* to unravel. Subsequently, given these traces, pertinent queries are generated. Finally, the *detective* is spawned in the environment, tasked with navigating these traces and actively probing the environment, all to derive answers through abductive reasoning.

Table 1: **Comparison between 🔍 Conan and related visual reasoning benchmarks.** 🔍 Conan is unique for its active reasoning and interactive multi-round setting on abductive reasoning tasks.

| Benchmark | Format | Multimodal | Interactive | Multi-round | Abductive |
|---|---|---|---|---|---|
| CLEVR (Johnson et al., 2017) | image | ✓ | ✗ | ✗ | ✗ |
| IQA (Gordon et al., 2018) | embodied | ✓ | ✓ | ✗ | ✗ |
| EmbodiedQA (Das et al., 2018) | embodied | ✓ | ✓ | ✗ | ✗ |
| $\mathcal{ART}$ (Bhagavatula et al., 2019) | language | ✗ | ✗ | ✗ | ✓ |
| VAR (Liang et al., 2022) | video | ✓ | ✗ | ✗ | ✓ |
| Sherlock (Hessel et al., 2022) | image | ✓ | ✗ | ✗ | ✓ |
| 🔍 Conan (Ours) | open-world | ✓ | ✓ | ✓ | ✓ |

## 3.1 Basic Components

**Playground** Originating from the Crafter playground, 🔍 Conan operates within a $64 \times 64$ grid matrix. Agents navigate this space with a localized $9 \times 9$ grid field of view centered on their current position. Once the *detective* is created in the environment, all traces left behind by the *vandal* persist, serving as clues for the *detective* to unravel. While pivotal studies (Johnson et al., 2016; Fan et al., 2022; Cai et al., 2023; Wang et al., 2023) address perception in 3D Minecraft settings using foundational models, our emphasis is on honing active abductive reasoning. To this end, we transition from a 3D visual perception to a 2D plane, ensuring a harmonious blend of reduced visual complexity and retaining rich interactivity (Xie et al., 2021).

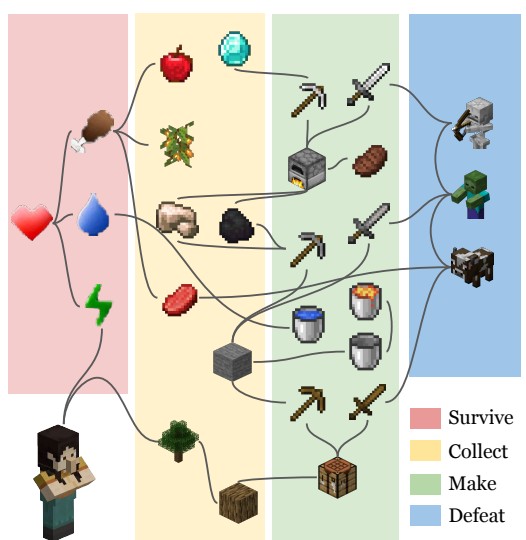

Figure 2: **Part of the task dependency graph**. Starting from the root note, any path forms a multi-step task for an agent to interact with the environment.

**Items and Actions** 🔍 Conan offers an extensive assortment of interactive items: food, materials, mobs, and tools, each tied to specific actions, as illustrated in Fig. 2. It furnishes 26 unique actions to foster agent-environment engagement. Certain actions leave traces, and together, the items and their mechanics provide a rich set of affordances for agents in the playground. This knowledge about item operations and traces aids the *detective* in comprehending the incident scene. Advancing from its predecessor, the original Crafter, 🔍 Conan now boasts over 30 achievements, a significant rise of over 50%. It features 32 distinct traces covering all agent actions such as crafting, collecting, defeating, eating, drinking, and incurring injuries. This enhancement enables the design of 60 varied abductive reasoning tasks within the scene. For an in-depth overview of the playground, refer to Appx. A.

*Vandal* Each 🔍 Conan map starts with the initialization of a *vandal*. This agent is driven by two primary aims: executing a specific task and preserving its existence within the environment. It is noteworthy that external threats might terminate the *vandal* prematurely. Traces left in the aftermath of the *vandal*'s activities form the question foundation for the *detective*, with every trace potentially birthing several questions. For a detailed overview, see Sec. 3.2. We model the *vandal* as optimal: when given a random task and the full map, it strategically delineates a sequence of subgoals based on the task dependency graph, all while ensuring its survival. In scenarios with multiple viable paths to an objective, uniform sampling comes into play. This sampling, supported by a probabilistic parser, presents varied strategies for task completion. Hence, the *detective* must delve deeper to distinguish the actual sequence of events from possible decoys. The execution of the *vandal*'s individual actions, as per the planned subgoal sequence, is steered by a collection of pre-established policies.

*Detective* After generating questions from a given trace, a *detective* is spawned to answer them. Traces left by the *vandal* span multiple steps and are only partially observable within the *detective*'s $9 \times 9$ grid field of view. This requires the *detective* to actively interact with the environment and gather evidence to answer the questions.

Though both *detective* and *vandal* share the same action space, the *detective* boasts a unique capability. It not only navigates and interacts like the *vandal*, but can also generate its own traces during its investigation. These overlaid traces from the *detective* enhance the environment's depth and complexity. This setup pushes the agent to actively derive conclusions from its dynamic interactions. Importantly, the *detective* is invulnerable; its focus lies squarely on problem-solving, eliminating concerns about survival or evasion. This design emphasizes active exploration and reasoning, ensuring 🔍 **Conan**'s primary goal remains addressing complex reasoning tasks and answering visual scene-related questions.

## 3.2 Questions and Choices

🔍 **Conan** is designed to assess the abductive reasoning capability of machine models through a diverse set of questions varying in difficulty and abstraction. These questions fall into three primary categories: **Intent** (local intent), **Goal** (global goal), and **Survival** (agent's survival status change). We approach evaluation as a multi-choice question-answering task. Each question offers four choices, with only one being correct. Questions and choices derive from predefined templates, as showcased in Tab. 2. For a more detailed explanation, see Appx. B.1.

Table 2: **Examples of three categories of questions in 🔍 Conan created from predefined templates.**

| Type | Questions |
|------|-----------|
| Intent | What did the *vandal* make on this table?
A: wood sword; B: wood pickaxe; C: iron sword; D: stone sword;
Why did the *vandal* cut a tree here?
A: make table; B: make wood sword; C: make finance; D: collect apple; |
| Goal | What was the *vandal*'s primary objective in this scenario?
A: get diamond; B: defeat zombie; C: collect apple; D: make iron sword;
What was the desired outcome of the task performed by the *vandal*?
A: make steak; B: make table; C: defeat skeleton; D: collect lava; |
| Survival | Why did the *vandal* die in this situation?
A: lack of water; B: lack of food; C: hurt by monster; D: hurt by lava;
What could the *vandal* have done differently to avoid a negative outcome?
A: avoid monsters; B: get sleep; C: get food; D: get water; |

**Intent** questions target the *vandal*'s immediate objectives or intentions during its task. To decipher these traces, agents must deduce the *vandal*'s underlying intent or subgoals. Solving these questions necessitates a learning model's comprehension of the local context.

**Goal** questions probe the *vandal*'s overarching objectives, extending beyond immediate intents. They necessitate grasping the wider context of a task or action sequence. Such questions query the *vandal*'s ultimate aims, demanding a learning model to reason within the broader context of the traces.

**Survival** questions address the wider investigative scope, posing added challenges to the *detective*. Centered on the *vandal*'s survival status changes during tasks (*e.g.*, collecting food for sustenance), they lead to deviations from the optimal action plan. While not tied to a task's primary objective, these questions require a deeper grasp of the present context, often necessitating reasoning around potential scenarios or alternate results.

Compared with the prevalent VQA setup, wherein questions are based on factual information that is readily obtainable from the input, 🔍 **Conan** questions cannot be deciphered given only the initial information, necessitating further exploration in the scene. Unlike standard embodied question answering, 🔍 **Conan** questions cannot be directly parsed as modular primitives; they demand abductive reasoning, drawing from both new observation and former knowledge to hypothesize, validate, and revise. For benchmarking purposes, 🔍 **Conan** produced a corpus comprising 100,000 questions. These were derived from 10,000 unique scenes, generated via the Crafter's scene generator, with each scene stemming from a task executed by a *vandal*. This resulted in an average generation of 10 questions per scene.

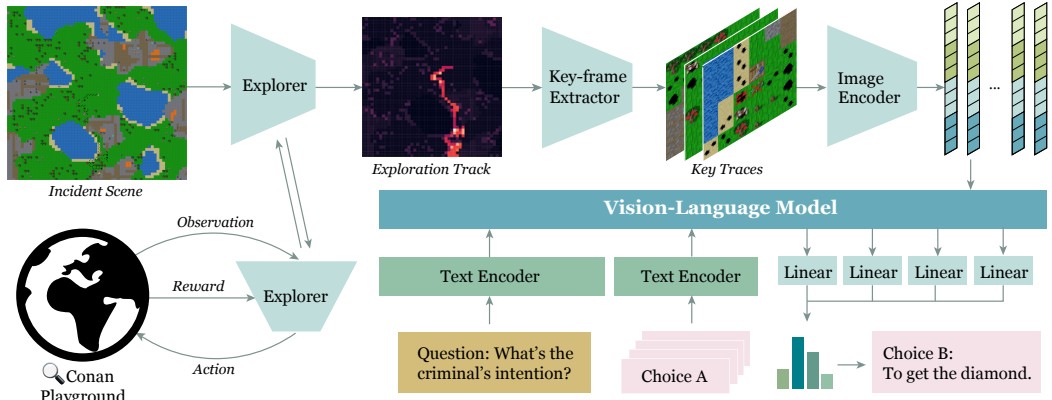

Figure 3: **An illustration of the detective pipeline for 🔍Conan.** An RL explorer is first trained to gather traces in accordance with the given question. Given a question and the incident scene, the *detective* calls the explorer subroutine to gather evidence. Next, the exploration sequence undergoes key-frame extraction, processed by a visual encoder, subsequently feeding into a vision-language model for answer selection.

# 4 The Detective Pipeline

🔍Conan casts the abductive reasoning challenge as a detective game, necessitating a *detective* to efficiently explore and gather information from the environment to deduce plausible explanations (*i.e.*, answers) for the given question. This process involves taking into account the temporal dependencies and incompleteness of the traces. To tackle these challenges encountered in 🔍Conan, we devise a detective pipeline, as depicted in Fig. 3.

Building on previous work that utilizes hierarchical models for task decomposition (Gordon et al., 2018; Das et al., 2018; Wijmans et al., 2019), our pipeline is structured into two primary phases: an exploration phase for trace collection, followed by an abductive reasoning phase. Initially, interaction with the playground is carried out to collect relevant visual information, which is subsequently leveraged in the reasoning phase to infer answers to the posed questions.

Computationally, our pipeline first employs RL agents as *explorers* (see Sec. 4.1) that learn an exploration policy based on the traces and the question, thereby rendering it goal-oriented. Next, given the question, we recruit vision-language models (see Sec. 4.3) to predict the answer based on the observation. A *key-frame extractor* (see Sec. 4.2) is inserted into the two phases to selectively identify relevant frames for abduction. The individual components undergo separate training procedures.

## 4.1 Explorer for Trace Gathering

The primary responsibility of an explorer is to efficiently collect information pertinent to the provided question. Initially, masks are employed to encode questions by highlighting relevant grids. Subsequently, the explorer takes in both the observation and the target question as input and outputs the action probability.

We use a reward function that incentivizes the agent to scout for clues and traces relevant to the given question. Additionally, a penalty term is incorporated to discourage unnecessary actions and inefficient searching, thereby promoting a more targeted exploration strategy.

Specifically, the agent is rewarded with $+1$ when a trace first appears within its local view, or $+2$ when the trace bears a close association with the question. A substantial reward of $+100$ is conferred upon the agent if it successfully uncovers all traces left by the *vandal*. Concurrently, the agent incurs a penalty of $-0.1$ for every timestep elapsed, with an additional penalty of $-1$ imposed for executing operating actions.

We evaluate multiple well-regarded RL frameworks as our explorer, including Deep Q-Network (DQN) (Mnih et al., 2015), Trust Region Policy Optimization (TRPO) (Schulman et al., 2015), and Recurrent Proximal Policy Optimization (RecurrentPPO) (Schulman et al., 2017). The Stable-Baselines3 library (Raffin et al., 2021) is employed for all implementations.

## 4.2 Key-Frame Extractor

Given that the frames gathered by the explorer tend to be excessively lengthy and redundant, a key-frame extractor is utilized to sift through and select informative frames containing crucial evidence for the *detective*. We adopt a prevalent selection strategy employed in video understanding (Arnab et al., 2021). Specifically, frames within the temporal bounds determined by the detection of the first and last traces are retained, from which $k$ frames are uniformly sampled. This design is intended to tailor the input with the constrained context window size to downstream vision-language models.

## 4.3 Vision-Language Models for Abductive Reasoning

We employ a multi-choice question-answering paradigm akin to the one used in Ding et al. (2021). Specifically, the model is presented with a question, its corresponding exploration frame sequence, and each potential answer choice, subsequently generating a score for each choice. The model is trained with a categorical cross-entropy loss. During inference, the choice with the highest score is considered the answer. We evaluate several well-established multimodal models; these models are known for their efficacy in processing both visual and textual data. Additional details on model implementation can be found in Appx. D.1.

**Vanilla-Trans**  The first baseline method leverages a vanilla transformer encoder to fuse observation and textual inputs. Specifically,the raw symbolic map from 🔍 **Conan** serves as the visual feature, while CLIP's text encoder (Radford et al., 2021) is employed to encode the textual input.

**FrozenBiLM**  FrozenBiLM (Yang et al., 2022), a state-of-the-art model for video question answering, combines visual input with frozen bidirectional language models, trained on web-scraped multimodal data. The approach integrates a frozen language model and a frozen vision encoder with light trainable visual projection modules. FrozenBiLM is tested with BERT-Large (Kenton and Toutanova, 2019) and DeBERTa-v3 (He et al., 2022) as the language model within our question-answering system, utilizing the symbolic map from 🔍 **Conan** for visual input.

**Flamingo-Mini**  Flamingo (Alayrac et al., 2022) is a family of vision-language models adept at rapid adaptation to novel tasks with minimal annotated examples. These models can handle sequences of visual and textual data, seamlessly accommodating interleaved images or videos as input. We finetune an open-sourced Flamingo-Mini model with frozen OPT-125M (Zhang et al., 2022), utilizing the symbolic map from 🔍 **Conan** for visual input.

## 4.4 Abduction from Deduction (AfD)

The adage "*Set a thief to catch a thief*" suggests the use of someone with a similar background or expertise to apprehend a wrongdoer: the best vandal catchers are vandals. This notion resonates with the core principle of Abduction from Deduction (AfD): for a skillful *detective* to abduce what a *vandal* does, it needs an in-depth grasp of vandals' *modus operandi*, motivations, and decision-making process. Translating the implication to a mathematical language, we articulate the problem of abductive reasoning based on evidence and knowledge from known deductive transitions. It can also be seen as an extension of inverse planning (Baker et al., 2007, 2009; Baker and Tenenbaum, 2014). Formally, let $g$ denote the goal of the *vandal*, $O$ the *detective*'s observation, and $S$ the playground states post the *vandal*'s actions. We then have:

$$P(g \mid O) = \mathop{\mathbb{E}}_{P(S|O)}[P(g \mid S,O)] = \mathop{\mathbb{E}}_{P(S|O)}[P(g \mid S)], \tag{1}$$

where we assume the independence of $g$ w.r.t. $O$ given $S$, as the goal ought to be clear given the states. Leveraging Bayesian rules, we further observe that

$$P(g \mid S) \propto P(S \mid g) \propto \prod_i \pi(a_i \mid s_i, g), \tag{2}$$

assuming a uniform prior over $g$ and known deterministic environment transitions. Eq. (2) asserts that $P(g \mid S)$ is proportional to a goal-conditioned forward action policy, where $s_i, a_i \rightarrow s_{i+1}$.

Intuitively, Eqs. (1) and (2) can be understood as follows: to **abduce** the *vandal*'s goal from observation, it is imperative to first reconstruct the actual states traversed by the *vandal* and subsequently ascertain the most plausible goal that, if pursued forward, would result in those states; see Eq. (1). Eq. (2) can be interpreted as a form of **deduction**, being contingent on transition knowledge derived from a forward action policy. Hence the name Abduction from Deduction (AfD).

In practice, two approaches emerge for implementing $P(g \mid S)$ based on Eq. (2). The first entails iterating over all $g$ and utilizing a learned or predefined $\pi(\cdot)$ to score a lengthy sequence of states. Conversely, the second approach embraces a data-driven strategy, wherein one arbitrarily selects $g$, samples $S$ from $\pi(\cdot)$, and learns a model of $P(g \mid S)$ using the $(g, S)$ pairs. The former approach proves time-intensive during inference due to the combinatorial temporal space and expansive goal space, thereby compelling us towards the latter approach. For implementation, we train $P(S \mid O)$ independently as a Dirac delta function of $\delta(f(O))$ and $P(g \mid S)$ from sampled pairs from $\pi(\cdot)$ employed in task execution in the *vandal*. The derived goal features, along with the question, are fed into the model for answer prediction. Please refer to Appx. F for additional details.

## 5 Experiments

### 5.1 Experimental Setup

**Exploration**  The explorer is trained using DQN, TRPO, and RecurrentPPO for $10^8$ steps, with a buffer size of $10^7$ and a batch size of $512$. In the case of DQN, training is conducted with $\epsilon = 0.96$. Each episode is capped at a maximum of $500$ steps for the explorer. A curriculum is employed to encourage long-term exploration whilst maintaining a balance with local search: initial training is carried out with traces from long-horizon tasks like "get the diamond," compelling the agent to venture farther from its starting point. Subsequently, the agent undergoes further finetuning across the entire dataset. Such a curriculum design prevents a sole focus on local discovery. For downstream reasoning models, $k = 30$ keyframes are extracted by the key-frame extractor.

**Abductive Inference**  Our reasoning models are tested under three different settings: Standard, Ideal Explorer, and AfD. In the Standard setting, models undergo training and testing based on the explorer's exploration. The Ideal Explorer setting sees models leveraging on an optimal exploration policy—visible to the ground-truth *vandal*'s trajectory, albeit imperfect, it facilitates the agent in gathering sufficient evidence for reasoning. This scenario can be conceived as a measure of the reasoning model's aptitude for passive reasoning given complete information. Under the AfD setting, models are trained and used as delineated in Sec. 4.4. All models are trained utilizing 8 NVIDIA GeForce RTX 3090 GPUs. For further training specifics, please refer to Appx. D.2.

### 5.2 Results and Analysis

Fig. 4 shows the learning curves of various RL agents during exploration. TRPO and RecurrentPPO manifest similar performance in terms of rewards following a substantial number of steps, markedly surpassing the DQN explorer. Additionally, we probe the impact of augmenting the maximum number of exploration steps to $5,000$ on performance. The data suggests a marginal performance uplift. Nonetheless, we acknowledge that such a performance increment is at the expense of substantially longer exploration time and a notable surge in the accrual of unrelated information. Consequently, we select TRPO with a maximum of $500$ steps per episode as our standard RL explorer.

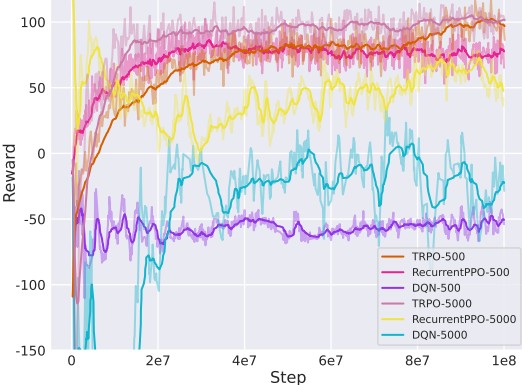

Figure 4: **Learning curves of various RL explorers.** The suffix $n$ denotes the maximum number of steps per episode during exploration. Results show that (i) TRPO and RecurrentPPO markedly outperform DQN in performance, and (ii) longer episodes marginally contribute to the performance at the expense of longer exploration time and the accrual of unrelated information.

Quantitative results on 🔍 **Conan** are depicted in Tab. 3; both the standard and AfD results reported employ TRPO as the explorer. In the standard setting, we discern that while models exhibit some aptitude in tackling low-level Intent questions, they struggle with higher-level questions pertaining to Goal and Survival. Among the models, Flamingo-Mini ascends to the pinnacle with an accuracy of $66.3\%$. FrozenBiLM models also perform relatively well. Notably, the DeBERTa variant slightly outperforms BERT, insinuating that a robust language backbone can improve general comprehension. Contrarily, the Vanilla-Trans model languishes across all tasks, achieving merely random-level performance.

Table 3: **Performance of abductive reasoning models on 🔍 `Conan`.** We report the question-answering accuracy (%) across various settings, with the overall accuracy averages over all question categories. F-BiLM refers to the FrozenBiLM model. **I** denotes Intent, **G** denotes Goal, **S** denotes Survival, and **O** denotes Overall. Results exhibiting the top individual performance are highlighted in **bold**, while models with the superior overall performance are shaded in gray.

| | Standard | | | | Ideal Explorer | | | | AfD | | | |
|---|---|---|---|---|---|---|---|---|---|---|---|---|
| | I | G | S | O | I | G | S | O | I | G | S | O |
| Vanilla-Trans | 32.9 | 25.0 | 24.5 | 28.8 | 64.0 | **78.4** | 58.1 | 66.1 | 24.8 | 23.3 | 24.5 | 24.3 |
| F-BiLM-BERT | 72.6 | **44.4** | **54.4** | 61.0 | 87.5 | 59.5 | 61.5 | 74.0 | 82.8 | **42.9** | **55.5** | 66.0 |
| F-BiLM-DeBERTa | 82.9 | 43.1 | 52.2 | 65.3 | **87.7** | 71.8 | **63.9** | **77.8** | 82.9 | 41.9 | 53.8 | 65.4 |
| Flamingo-Mini | **86.2** | 43.3 | 49.5 | **66.3** | 85.8 | 47.8 | 56.6 | 69.0 | **84.9** | 42.5 | 52.2 | **66.1** |

With the Ideal Explorer, we notice a clear performance boost across all tasks, particularly in the Goal and Survival questions. These results allude to the potential bottlenecking of models' abductive reasoning capability due to the insufficient information collected, underscoring the significance of effective exploration. An adept explorer can significantly aid in the accrual of useful information, informatively pursuing a hypothesis to scrutinize evidence, swiftly self-correcting upon encountering conflicting evidence, and reasonably re-planning. The findings also hint sufficient room for the RL explorer to improve. Remarkably, the Vanilla-Trans exhibits the greatest increase, insinuating that, in comparison to other baseline models, it is markedly vulnerable to insufficient evidence.

For AfD results, nearly all multimodal models exhibit performance on par with end-to-end supervisedly trained models. Remarkably, FrozenBiLM models even surpass the performance observed in standard settings. The persisting failure of Vanilla-Trans can be ascribed to its weakness in reasoning amidst incomplete observations due to the significant disparity between the familiar complete state $S$ and incomplete observation $O$. Examining task-specific results, a notable performance uplift in the Survival task models is discernible for almost all models relative to the standard setting, albeit sharing the same observation. These results intimate that the inclusion of deductive information sensitizes the *detective* to *vandal*'s concerns during task execution. Nevertheless, the exhibited performance in long-term planning remains weak, reinforcing the pressing need for a better exploration policy. Critically, these models continue to find short-term intent questions to be most easily answered.

## 5.3 Further Discussion

**Additional Experiments** We further experiment in the absence of visual inputs, serving as a negative control baseline, resulting in random performance across all settings; see Appx. E. This random-level performance underscores the severe constraints imposed on the agent without visual information. The TRPO explorer shows a noticeable improvement over the ones without visual inputs, suggesting that even minimal exploration is preferable to none. Nonetheless, the performance remains relatively modest. On the other hand, the Ideal Explorer demonstrates markedly superior performance, attesting to the substantial benefits its capacity to accrue perfect trace evidence renders to the downstream reasoning task. This accentuates the imperative of effective exploration.

Table 4: **Error analysis on 🔍 `Conan`.** We examine the accuracy of FrozenBiLM-DeBERTa across various tasks, comparing two explorer groups: reasoning based on the TRPO explorer and the Ideal explorer (in gray).

| get_drink | defeat_cow | get_apple | defeat_skeleton | make_iron_pickaxe |
|---|---|---|---|---|
| 47.06 | 43.90 | 35.7 | 46.59 | 56.52 |
| 100.00 | 85.37 | 78.57 | 82.95 | 52.17 |

| place_bed | make_steak | make_stone_pickaxe | get_coal | make_stone_sword |
|---|---|---|---|---|
| 43.90 | 46.15 | 48.48 | 50.00 | 37.50 |
| 87.80 | 50.00 | 39.39 | 45.45 | 4.17 |

| get_iron | get_water | get_stone | make_iron_sword | place_furnace |
|---|---|---|---|---|
| 28.57 | 45.95 | 36.84 | 56.25 | 44.44 |
| 46.43 | 54.05 | 47.37 | 28.12 | 83.95 |

| get_diamond | place_table | get_wood | make_wood_pickaxe | make_wood_sword |
|---|---|---|---|---|
| 40.62 | 39.36 | 36.00 | 40.00 | 50.00 |
| 84.38 | 91.49 | 96.00 | 55.00 | 64.29 |

| make_bed | get_lava | make_bucket | get_beef | defeat_zombie |
|---|---|---|---|---|
| 47.83 | 50.00 | 35.29 | 53.85 | 52.50 |
| 39.13 | 66.67 | 73.53 | 42.31 | 75.00 |

**Error Analysis**    We extend an error analysis for the "goal" split, probing the reasoning model across a spectrum of tasks. Table 4 compares two groups: reasoning based on the Ideal explorer and the TRPO explorer. The findings underscore that proficient exploration, *i.e.*, the heuristic Ideal explorer who recovers the vandal's trajectory, is sufficient for satisfactory performance. However, to fully harness the potential, a more adept reasoner is requisite, one capable of deciphering the vandal's hidden states from observed traces. For instance, the act of felling trees could signify a need for either wood or food (apples), and discerning the intent solely from traces of felled trees presents a challenge. When it comes to "trace-relevant" frames or "keyframes," the Ideal explorer could ostensibly furnish all trace-relevant frames. However, the concept of keyframes remains nebulous. Within the video understanding domain, a formidable challenge lies in the extraction of "keyframes." This is a post-hoc concept that eludes straightforward acquisition upfront. A prevailing approach, aimed at augmenting efficiency (diminishing context length in Transformer), entails truncating it via every k-th frame.

**Joint Reasoning**    The collective enhancement of both exploration and reasoning elements emerges as quintessential, given its mirroring of human-like intelligence. For instance, by providing feedback, the reasoner can steer the explorer towards actions that are potentially more insightful and likely to produce pertinent traces. Nonetheless, practical implementation encounters significant hurdles. Assigning credit to exploratory decisions bearing long-term implications can be intricate, particularly when the outcomes of exploratory actions become evident after a substantial time lapse, thereby muddying the causal relationship between the decisions and their ultimate effect on reasoning and answering questions. This accentuates the mutual reliance between exploration and reasoning—advancement in one facet demands progression in the other, introducing a bilateral dependency that complicates optimization. The reasoning component alone demands hefty training and computational resources, especially when utilizing large language models. The demand for formidable computational power renders the simultaneous optimization of exploration and reasoning exceedingly daunting. Collectively, this approach is also widely adopted (Gordon et al., 2018; Lei et al., 2018; Kočiský et al., 2018). Consequently, we navigate along this trajectory, projecting that future endeavors on 🔍 `Conan` should prioritize reasoning above exploration.

To summarize, the engagement of a proficient explorer substantially enhances abductive reasoning, particularly in higher-level tasks such as goal-oriented and survival-centric inquiries. This underlines the criticality of exploration as a precursor to tackling abductive reasoning tasks in the presence of incomplete information. Furthermore, the achievement of the AfD hint at the potential for models to harness world knowledge, especially transition knowledge pertaining to tasks and traces, to transform abductive reasoning into deductive simulation. We posit that the presented approach resonates more with human-like reasoning, edging us closer to the core of human intelligence.

## 6    Conclusion

In this paper, we introduce 🔍 `Conan`, a benchmark tailored to evaluate and assess models' active reasoning ability in addressing incomplete-information questions in an interactive environment. 🔍 `Conan` sets itself apart from existing abductive reasoning benchmarks by incorporating an open-world playground facilitating active exploration. It differentiates itself from prevailing embodied question-answering benchmarks by introducing the demanding abductive process in question answering, necessitating multi-round abductive inference based on gathered evidence. Moreover, we propose a new learning paradigm, Abduction from Deduction (AfD), that turns the problem of abduction to deduction, exploiting the problem structure through Bayesian principles. Benchmarking the efficacy of contemporary machine learning models on 🔍 `Conan`, we elucidate the model limitations in interacting with the environment that leads to failure in higher-level, longer-term abductive reasoning.

**Limitations and Future Work**    In general, we notice two significant limitations from the experimental results. For one, the explorer does not supply particularly relevant information for the reasoning model. In the human abductive reasoning process, exploration and reasoning should be closely intertwined, with an agent using the current hypothesis to guide exploration and improve its understanding. However, due to long-range exploration and complex vision-language reasoning, we only applied the conventional visual question-answering method and did not fully integrate these two processes. For another, learning naive question-answer mapping shall be sub-optimal. By leveraging the problem structure, AfD has shown improved performance on a particular set of problems. Nevertheless, the current AfD formulation is still rudimentary. We believe an in-depth understanding of the structure and well-crafted implementation could further boost performance.

**Acknowledgement** The authors would like to thank Ms. Zhen Chen (BIGAI) for designing the figures, and NVIDIA for their generous support of GPUs and hardware. M.X., G.J., W.L., C.Z., and Y.Z. are supported in part by the National Key R&D Program of China (2022ZD0114900), M.X. and W.L. are supported in part by the NSFC (62172043), and Y.Z. is in part by the Beijing Nova Program.

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

# A 🔍 Conan Playground

🔍 Conan's playground is a computationally efficient 2D open-world environment with diverse items and rich tasks. The most distinctive feature of 🔍 Conan's playground over the original Crafter environment is that agents in 🔍 Conan leave diverse traces when interacting with the environment. These traces serve as the foundation for abductive reasoning; the *detective* has to effectively connect the traces to figure out what the *vandal* has done.

## A.1 Items and Traces

**Land**    Based on Crafter, there are three types of terrains that agents can walk on: *sand*, *grass*, and *path*. *Sand* and *grass* are soft surfaces where agents leave directional footprints after walking on them (see Fig. A1 first 2 rows in Columns 2 and 3 for examples). If a grid is left with more than one footprint, the footprints will become melded (Fig. A1 Column 4 in first 2 rows). Agents' actions will also leave traces on the terrain, *e.g.*, water on the ground (Fig. A1 Column 5 first 2 rows). If an agent gets injured, blood will be shed on the ground (Fig. A1 Column 6 first 2 rows).

**Creatures**    There are four creatures in the playground: *plant*, *cow*, *zombie* and *skeleton*. *plant* grows from sapling to ripe plant. *Cow* randomly wander on the ground, whereas *zombie* and *skeleton* (monsters in general) will target agents in sight: *zombie* chases agents and *skeleton* shoots *arrow* at agents. Agents can fight with creatures and kill them. These actions will leave monster bodies on the ground.

**Tools**    Agents can make tools on the *table*. There are 7 tools in total: *bucket*, *wood_sword*, *wood_pickaxe*, *stone_sword*, *stone_pickaxe*, *iron_sword*, and *iron_pickaxe*. These tools can be made using different materials and used for certain tasks. Both swords and pickaxes can be used to fight with creatures, but only pickaxes can be used in mining. Buckets can be used to collect water and lava.

**Actions**    🔍 Conan's playground enables agents to interact with objects, non-playable characters, and even other agents in the playground. Agents can cut *tree* to get *apple* and *wood*, as well as collect *sapling* and grow *plant* (Fig. A1 Row 3). They can also mine with different tools to get *stone*, *coal*, *iron*, and *diamond*. Using these materials, agents can make *bed* for sleep, *furnace* for keeping monsters away and grilling food, *table* for making tools, *etc*. Of note, these items should be placed in an empty grid to use and they can be destroyed by monsters.

## A.2 Achievements and Tasks

There are 60 tasks and 39 achievements in 🔍 Conan's playground. We list all achievements in Tab. A1. Tasks are composed achievements. We select 60 nontrivial and meaningful tasks from all compositions in 🔍 Conan as the final task set.

Table A1: **Achievements in 🔍 Conan.**

| Type | Achievements | | | |
|---|---|---|---|---|
| Survive | drink_water
sleep
drink_water_from_bucket | eat_apple
sleep_on_bed
eat_plant | eat_beef
wake_up | eat_steak
eat_grilled_apple |
| Collect | collect_wood
collect_iron
collect_water | collect_apple
collect_diamond
collect_lava | collect_water
collect_beef
collect_sapling | collect_stone
collect_coal
collect_plant |
| Make | make_steak
make_wood_sword
make_iron_sword
place_furnace | make_grilled_apple
make_wood_pickaxe
make_iron_pickaxe
place_plant | make_bucket
make_stone_sword
place_table | make_fence
make_stone_pickaxe
place bed |
| Defeat | defeat_cow | defeat_zombie | defeat_skeleton | |

## A.3 Observation and Action

🔍 Conan offers both pixel representation and symbolic representation for training agents. For pixel representation, the environment returns a $900 \times 900$ RGB image each time step for the *detective*'s

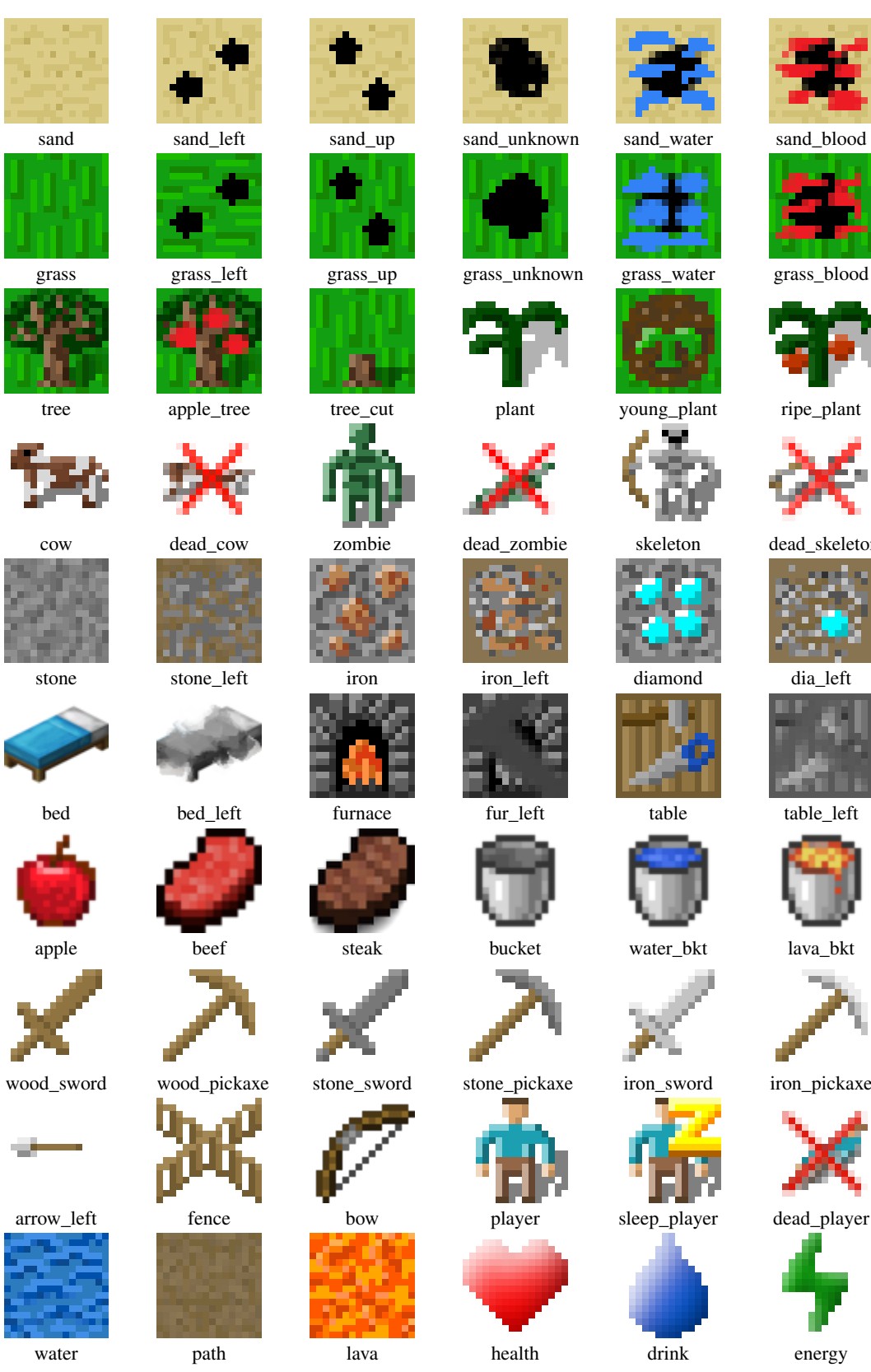

Figure A1: **Items and related traces in** 🔍 **Conan.**

$9 \times 9$ local view. For symbolic representation, the environment returns a $9 \times 9$ tensor, with each entry an index representing one of 50 grid types, covering materials, resources, objects, creatures, and *etc*. The agent is always at the center of the observation.

🔍 **Conan** affords a larger action space. See Tab. A2 for a detailed list of actions.

Table A2: **Actions in** 🔍 **Conan.**

| Action | Details |
| --- | --- |
| Noop | Do nothing. |
| Move Left | Move left if the grid is walkable. |
| Move Right | Move right if the grid is walkable. |
| Move Up | Move up if the grid is walkable. |
| Move Down | Move down if the grid is walkable. |
| Do | Collect materials or fight with monsters. Use tools if possible. |
| Sleep | Sleep to restore energy. Sleep on bed can restore energy faster; |
| Place Stone | Place a stone if the grid is not occupied. Should have a stone. |
| Place Table | Place a table if the grid is not occupied. Should have a table. |
| Place Furnace | Place a furnace if the grid is not occupied. Should have furnace. |
| Place Plant | Place a plant if the grid is grass. Should have sapling. |
| Place Bed | Place a bed if the grid is not occupied. Should have bed. |
| Make Wood Pickaxe | Nearby table. Should have wood. |
| Make Stone Pickaxe | Nearby table. Should have wood, stone. |
| Make Iron Pickaxe | Nearby table, furnace. Should have wood, coal, iron. |
| Make Wood Sword | Nearby table. Should have wood. |
| Make Stone Sword | Nearby table. Should have wood, stone. |
| Make Iron Sword | Nearby table, furnace. Should have wood, coal, iron. |
| Make Bucket | Nearby table. Should have wood, stone. |
| Make Steak | Nearby table, furnace. Should have beef. |
| Eat Apple | Restore 2 health. Should have apple. |
| Eat Beef | Restore 4 health. Should have beef. |
| Eat Steak | Restore 6 health. Should have steak. |
| Collect Water | Collect water to bucket. Should have empty bucket. |
| Collect Lava | Collect lava to bucket. Should have empty bucket. |
| Drink | Drink water. Drink water from water bucket if not near the water. |

## B 🔍 Conan Questions

### B.1 Question Generation

Questions in 🔍 **Conan** are generated based on *vandal*'s task-finishing process. To generate a question, (1) we initialize a playground and put the *vandal* in it; (2) the *vandal* is randomly assigned a task; (3) the *vandal* tries to finish the task with the help of the pre-build parser and planner, and generates logs along the way; (4) a question is generated based on a certain part of the log. We randomly select a template from the template pool and fill placeholders with related objects in it. The answer is also parsed from the log. Other choices are sampled based on the question and the context to avoid unrelated choices that can be easily excluded.

### B.2 Question Templates

Tab. A3 lists all the templates we use for generating questions.

### B.3 Dataset Statistics

See Tab. A4 and Tab. A5 for details.

## C Explorer

The Explorer in the *detective* is an RL agent. The agent receives an observation of a $[64, 64, 2]$ tensor. This tensor combines the $9 \times 9$ symbolic local view of the *detective* and a $64 \times 64$ question mask.

Table A3: **Question templates in 🔍 Conan.** [] is the placeholder.

| Type | Templates | |
|---|---|---|
| Intent | What was the *vandal*'s objective in these area? | What was the *vandal*'s current intent? |
| | What did the *vandal* do after this step? | What did the *vandal* do before this step? |
| | What did the *vandal* make on this table? | Why did the *vandal* make this table? |
| | What item did the *vandal* most likely craft using the table? | Why did the *vandal* make the []? |
| | What action did the [] perform immediately? | What was the [] used for? |
| | What did the *vandal* make on this furnace? | Why did the *vandal* make this furnace? |
| | What item did the *vandal* most likely craft using the furnace? | Why was tree cut? |
| | What was the intended use for the wood? | How was the tree cut? |
| | What was the purpose of mining []? | Why was the [] mined? |
| | What was the intended use for the []? | How did the *vandal* defeat the []? |
| | What did the *vandal* use to defeat the []? | Why did the *vandal* defeat the []? |
| Goal | What was the *vandal*'s final goal? | What was this *vandal* trying to achieve? |
| | What did the *vandal* want to achieve? | |
| Survival | What was the *vandal*'s survival intent for doing []? | why did the *vandal* collect/make []? |
| | What was the *vandal*'s goal for survival currently? | Did the *vandal* die? Why? |
| | Why did the *vandal* die during the task? | How did the *vandal* die? |
| | What was the *vandal* trying to do when died? | What can the *vandal* do to avoid death? |
| | what helped keep the *vandal* away from hungry? | what food did the *vandal* eat? |

Table A4: **Dataset split and choice distribution.**

| Category | Train | Test | Val | Choice A | Choice B | Choice C | Choice D |
|---|---|---|---|---|---|---|---|
| Intent | 71162 | 9152 | 8822 | 24.99% | 25.20% | 24.89% | 24.93% |
| Goal | 8000 | 1000 | 1000 | 24.89% | 25.08% | 24.87% | 25.16% |
| Suvival | 7365 | 1560 | 1596 | 25.13% | 24.95% | 24.95% | 24.97% |

Table A5: **Task distribution.**

| Task Percentage | get_drink | defeat_cow | get_apple | make_stone_pickaxe | place_bed | place_furnace |
|---|---|---|---|---|---|---|
| | 2.47 | 8.49 | 2.52 | 2.87 | 8.44 | 8.23 |
| get_lava | defeat_skeleton | make_iron_sword | get_coal | get_beef | get_diamond | get_stone |
| 2.72 | 8.7 | 2.64 | 2.42 | 2.7 | 2.39 | 2.67 |
| make_bucket | get_iron | get_water | make_iron_pickaxe | make_bed | make_steak | make_wood_sword |
| 3.11 | 2.44 | 2.2 | 2.95 | 2.71 | 2.81 | 2.53 |
| defeat_zombie | make_stone_sword | place_table | get_wood | make_wood_pickaxe | | |
| 7.8 | 2.65 | 8.24 | 2.67 | 2.63 | | |

The local view is zero-padded to $64 \times 64$. This ensures the agent knows its relative position on the map. Additionally, the mask is generated based on the question, with the area related to the question unmasked. The mask serves as the goal of the exploration policy.

All the RL baselines are trained for $10^8$ steps. See more details below. Unless specified otherwise, parameters are set as default in Stable Baselines.

### C.1 Model Details

**DQN** The DQN baseline is trained using a $\gamma$ value of 0.98, a $\tau$ value of 1, a learning rate of 0.0001, a buffer size of $10^7$, and a batch size of 512. We leverage an MLP policy with two layers of 64 neurons each. The model is updated every 10 steps.

**TRPO** The TRPO baseline updates its policy with a special KL-divergence constraint on the distance between the new and old policies. We also leverage an MLP policy for TRPO, where the same multi-layer perceptron is used for both policy and value prediction.

**RecurrentPPO** The ReucrrentPPO baseline uses long short-term memory (LSTM) (Hochreiter and Schmidhuber, 1997) as the recurrent policy. The LSTM layers' weights are initialized with standard Gaussian. We reset LSTM states at the end of the episode. The LSTMs for both the actor and the critic have the same architecture, with two LSTM layers of 256 neurons each.

## C.2 Training Details

Explorers are firstly trained on long-horizon tasks as explained in the main text. These long-horizon tasks include "get diamond," "get lava," "get water," "make iron sword," "make iron pickaxe" and "eat steak." These tasks can be further broken down into over 20 subtasks and have an average episode length of more than 200 steps. We generate 10,000 unique scenes with traces given these tasks and train explorers on them for $10^8$ steps. Then the explores are fine-tuned on all tasks in 🔍 **Conan** for $10^7$ steps.

We also show the frame rate per second (FPS) for different RL baselines during training in Fig. A2. As can be seen from the figure, DQN exhibits the highest training efficiency, reaching an FPS exceeding 3000. TRPO maintains a stable FPS of 2000. On the contrary, RecurrentPPO operates significantly slower, requiring over 96 hours to complete training with 128 subproc environments, whereas TRPO accomplishes the task in just 14 hours.

# D  VL Reasoning

In this section, we describe the experimental details for the Vision-Language (VL) models used in the paper.

## D.1  Model Details

**Vanilla-Trans**  For Vanilla-Trans, the visual features together with the text features are concatenated in the format of `[frame_1, frame_2, ..., frame_n, question, choice_1, choice_2, ..., choice_4]`. Visual features, if from the symbolic observation, are directly passed into the model. Otherwise, we utilize CLIP's pre-trained image encoder

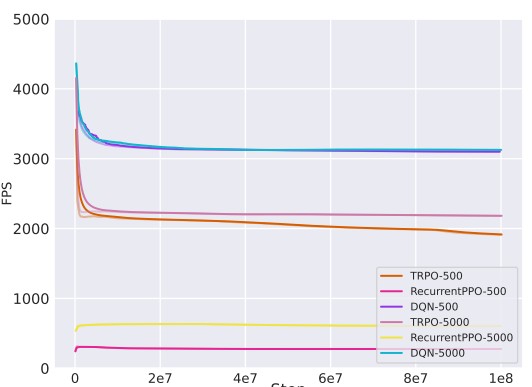

Figure A2: **Frame rate per second (FPS) curves of several RL explorers in training.** Results show that DQN and TRPO are significantly faster than RecurrentPPO.

(ViT-B/16) to extract features from pixel input. Text features are calculated using the text encoder of CLIP. These input features are then passed through a 6-layer Transformer model with an MLP head for classification.

**FrozenBiLM**  We adopt the cross-modal FrozenBiLM for 🔍 **Conan**, drawing inspiration from models used in Multiple-choice VideoQA benchmarks such as How2QA (Li et al., 2020) and TVQA (Lei et al., 2018)[1]. 🔍 **Conan** can be formulated as a multiple-choice VideoQA problem given the fixed explorer. We concatenate all of the observation frames as the video input. The questions and choices are converted into the following format: `["{question} Is it {choice_1}?," ..., "{question} Is it {choice_4}?"]`. We then evaluate the probabilities of the model producing "Yes" and "No". The visual features are processed in the same way as in Vanilla-Trans and then forwarded for visual-text projection. We utilize BERT-Large and DeBERTa as our frozen language backbones in this work; however, other general language models are applicable as well.

**Flamingo-Mini**  Our Flamingo-Mini baseline is based on an open-source implementation of the Flamingo model[2], as the original Flamingo model's pre-trained weights are not accessible. Flamingo-Mini is built upon OPT-125M and CLIP pre-trained ViT-L/14 model. We also formulate 🔍 **Conan** as a multiple-choice problem for Flamingo-Mini. The questions and choices are converted into the following format: `["Question: {question} Answer: {choice_1}," ..., "Question: {question} Answer: {choice_4}"]`. Each question-choice pair is fed into the model and then a binary classifier head is used on Flamingo's last layer output to predict the final answer.

---

[1] https://github.com/antoyang/FrozenBiLM
[2] https://github.com/lucidrains/flamingo-pytorch

## D.2 Training Details

Vanilla-Trans was trained for 100 epochs, with a batch size of 128. FrozenBiLM models were trained for 50 epochs, with a masking probability (for the MLM objective) of $0.15$, a batch size of 32, a learning rate of $3 \times 10^{-4}$, a gradient clipping max norm of $0.1$, and Adam as the optimizer ($\beta_1 = 0.9, \beta_2 = 0.95, \epsilon = 1 \times 10^{-8}$). Flamingo-Mini was trained for 100 epochs, with a learning rate of $5 \times 10^{-5}$, a batch size of 8, and also Adam as the optimizer ($\beta_1 = 0.9, \beta_2 = 0.999, \epsilon = 1 \times 10^{-8}$).

# E  Additional Experiments

## E.1  Negative Control Baselines

We compare our VL reasoning results on the trained explorers with those on empty visual inputs as a negative control baseline. The results are shown in Tab. A6.

Table A6: **VL Reasoning models' performance on explorers compared with empty visual inputs.**

|  | Vanilla-Trans | F-BiLM-BERT | F-BiLM-DeBERTa | Flamingo-mini |
|---|---|---|---|---|
| Empty visual inputs | 26.4 | 25.5 | 25.9 | 22.9 |
| TRPO explorer | 25.0 | 44.4 | 43.1 | 43.3 |
| Ideal explorer | 78.4 | 59.5 | 71.8 | 47.8 |

The results show that using empty visual inputs yields random performance across all settings. Besides, it also shows that the training QA pairs are unbiased. The TRPO explorer achieves higher performance, which suggests that the exploration strategy learned by TRPO helps gather some informative evidence for the reasoning process. The Ideal explorer is an oracle-like exploration policy that has access to perfect trace evidence and temporal information. It provides the most comprehensive information about the environment. This highlights the importance of effective exploration in improving reasoning performance. However, it does not mean that reasoning is less important, as even with the Ideal explorer, the model still could not achieve satisfactory performance. Based on all results, collecting informative evidence seems to be more important in the overall objective.

# F  Abduction from Deduction (AfD)

As mentioned in Sec. 4.4, we adopt a data-driven strategy to learn a model of $P(g \mid S)$ and simultaneously answer the questions. To be more specific, we train the *detective* agent self-supervisedly. The *detective* is randomly assigned with one of all possible tasks. It then finishes the task by following the action policy $\pi(\cdot)$. Note that we assume the *detective*'s $\pi(\cdot)$ is the same as the *vandal*'s in order to best implement the idea of AfD. Based on the task execution process, questions are generated. Since our ultimate goal is to have our models answer 🔍 **Conan**'s questions, we do not explicitly construct $P(g \mid S)$, but rather consider the question-answer process as the $g$. We then train $P(g \mid S)$, where $S$ is the *detective*'s observation during the task execution, and the label can be derived from the assigned tasks together with the $\pi(\cdot)$.

Besides $P(g \mid S)$, we still need to learn a model of $P(S \mid O)$, which, intuitively, can be understood as inferring the true state of the environment from partial observation. In our experiment, we tried two ways to model $P(S \mid O)$. One approach is to directly train a model using multi-frame observations to predict the states. We employed a UNet (Ronneberger et al., 2015) and a multi-layer CNN as the network. However, this method did not work effectively. Reasoning based on the reconstructed states only achieved performance at a random level. The second approach, which was finally used to report performance, aligned the hidden feature spaces from true states and observations. When training $P(g \mid S)$, we added a head before the VL models, converting the input $S$ into a 4096-dimensional vector. Then we trained a head on $O$ with the same structure, minimizing the difference between features from $O$ and features from $S$.

# G  🔍 Conan Task Demo

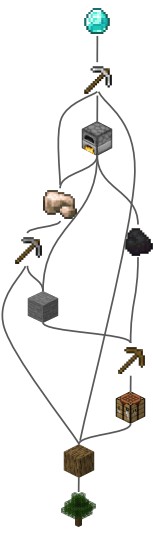

To better illustrate the core components in 🔍 **Conan**, We take the playground shown in Fig. 1 as an example. In this scenario, the assigned task is "get diamond" (Fig. A3 shows the task dependency). As shown in Fig. A4, once the vandal completes the task, it leaves behind traces in the playground. The vandal ends at the bottom of the figure. The detective then enters the playground, starting at the beginning of the traces. In this case, traces encompass footprints and remnants left after certain actions. Note that footprints cannot be left on sand or stone, and different footprints may overlap. The vandal will collect objects crafted on a table, making them invisible.

Let's suppose the detective's exploration begins by following footprints (note the context window size is 9×9).

Firstly we can see some cut trees. As the footprints are not seriously overlapped and mostly one-directional, we can deduce the vandal did not return. After seeing the tool-making table, with the only resources being wood, we could say that the vandal could only make wooden tools, not stone swords or iron pickaxes, further restricting possible actions the vandal took.

Note that this is already critical reasoning in 🔍 **Conan**.

Figure A3: **The task structure of "get diamond".**

Moving on, we note that footprints become missing on the sand surface. However, we note broken stones and coals. Therefore, the wooden tool to break stones and coals shall be a wooden pickaxe. So the agent should have made a wooden pickaxe on the table earlier. Despite the fact that the tool has been collected, we could still figure that out.

Following the reemerged footprints, we note blood and a zombie body on the ground, suggesting the vandal should have had a fight.

Searching on, we find the broken diamond. As an iron pickaxe is the only tool to collect diamonds. The vandal must have built an iron pickaxe with iron and coal in the furnace. With no other footprints around, we can safely conclude our search.

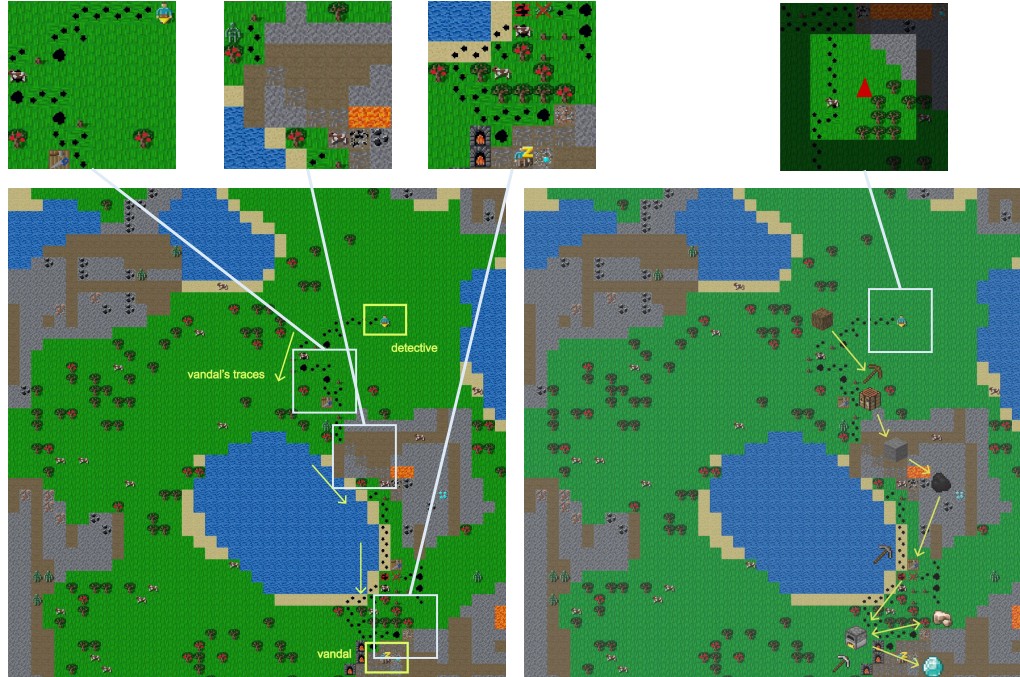

Figure A4: **A demostration of core components in 🔍 Conan.** We show how a *detective* can do reasoning based on the task structure and traces left in the playground. Zoom in for more details.

