# OpenReview forum: "Active Reasoning in an Open-World Environment"
_NeurIPS.cc/2023/Conference — NeurIPS 2023 poster_

### Official Review · Reviewer_yGBc · 2023-07-06

**Soundness:** 2 fair
**Presentation:** 2 fair
**Contribution:** 2 fair
**Rating:** 3
**Confidence:** 4

**Summary:**

This paper introduces Conan, a new benchmark for evaluating active reasoning in an embodied/open-world environment where an agent acts as a detective and must answer questions about the actions and intentions with the traces of another agent (a vandal). The experiments benchmark a RL + vision-and-language baseline and a more structured model that uses Bayesian inference on the Conan task.

**Strengths:**

- ambitious benchmark trying to tackle an exciting and interesting problem that highlights a limitation in the current focus of the community: lack of benchmarks testing reasoning/question answering with active information gathering
- problem and setting are well-motivated
- interesting setup for the active reasoning problem with a vandal/detective

**Weaknesses:**

- many important details that are not clear. One of the main things that needs more clarity is what the format of the benchmark is - it would be very helpful to have more examples and earlier in the paper of what the traces look like / what context is available to models, or maybe an example of a series of reasoning steps a model (or human) would go through to answer a question in Conan. See questions below for specific points of confusion
- I wasn't totally convinced/clear that the questions in Conan are focused on testing abductive reasoning. Although eventually we would want to solve this with end-to-end methods like the RL baselines presented here, I feel like some heuristic exploration baselines would be important to understand what it takes to solve the benchmark and contextualize the scores of the RL agents (what scores do these agents get?):
	- the ideal explorer (specifically wondering: is the ideal explorer the ideal policy?)
	- an agent that explores the whole 64x64 grid (how much better does the ideal explorer do compared to this?)
	- using an oracle extractor that extracts the trace-relevant frames instead of the every-k key-frame extractor
	- instead of using RecurrentPPO, some heuristic extractor that allows agents to condition on key frames in the past (e.g. the frames with traces that are closely associated with the question)
- Related to previous point, the benchmark also conflates *memory* and *exploration* with the abductive reasoning challenges -- both of these are already huge challenges for RL/current models. I worry that these may make the benchmark too difficult, and additionally not actually test for the reasoning abilities that it claims to test. Overall, I just don't have a good understanding of what solving the benchmark actually entails - if we have agents with good memory + exploration, is the reasoning involved actually quite simple? I think heuristic baselines/agents with privileged information would help the case here, as well as simplifying aspects of the task that are orthogonal to abductive reasoning

**Questions:**

- table 1: what's the difference between embodied and open-world?
- l146: what does a trace look like? are objects left behind, does the state change, etc.?. l151: when you say "32 distinctive traces" do you actually mean "32 object interactions" and traces can consists of multiple object interactions?
- l153: "60 abductive reasoning tasks" - does this mean 60 question templates?
- l164-168: what's an example of a path + alternative and why exactly does this require the detective to gather additional information?
- why is the detective allowed to take the same actions as the vandal instead of just exploring the scene and collecting information? (feels like potentially unnecessary complexity)
- l226-227: "masks are first employed to encode questions..." - what does this sentence mean?
- How does the explorer condition on the question (e.g. some sentence embedding of the question or something else?)
- Is there a step limit on the explorer's episodes?
- l174-177: this made it sound like the detective and the agent were separate. My understanding is that the agent *is* the detective?
- l255: what is the representation of symbolic map?
- l307: to clarify, does the ideal explorer get the frames of the ground truth vandal trajectory, and then we extract every 30 frames? how much of the gap in performance is due to the keyframe extractor dropping important frames vs. the inaccuracy of the reasoning model?
- sec 4.4: does this assume access to the vandal policy pi? Is that realistic?
- l345: why would vanilla-trans be more susceptible to insufficient evidence? Isn't a more likely explanation that it's not pretrained to do question answering, in contrast to the other models?
- AfD doesn't improve on standard (small improvements at best / mixed results across the board) - doesn't this suggest that this kind of reasoning is not super important in the benchmark?

**Limitations:**

Yes

---

> ### Author Rebuttal · Authors · 2023-08-10
>
> Dear reviewer:
>
> > "have more examples about traces and reasoning steps"
>
> See pdf attached.
>
> > "Are the questions in Conan focused on testing abductive reasoning? Some heuristic exploration baselines"
>
> Good exploration, ie, the heuristic ideal explorer recovers the vandal's trajectory, is enough for "OK" performance, but to sqeeuze out the juice, one needs much better reasoner that can figure out the hidden states from of the vandal, from observed traces (see response to nDEB). Eg, cutting trees could both mean the vandal is in need of wood or food (apples) and from only traces of cut trees, it is not easy to figure it out. For "trace-relevant" frames or "key frames", the ideal explorer could already be understood as providing all trace-relevant frames. The notion of key frames is ill-defined. As in the video understanding community, one major challenge is how to extract "key frames". This is a post-hoc concept that can not be easily obtained beforehand. The common wisdom, and to improve efficiency (reduce context length in Transformer) is to trunck it via every k frame. For the exhaustive search agent, we show the results with an agent with full 64x64 grid observation here: 25.6(Vanilla-Trans), 64.6(F-BiLM-BERT), 69.1(F-BiLM-DeBERTa), 67.9(Flamingo-mini). The noisy sequence, with unoptimal temporal order (unlike the ideal explorere), makes the results slightly worse.
>
> > "conflates memory and exploration with abductive reasoning"
>
> Memory and exploration are other terms to describe the challenge. However, they are not all. Indeed, the agent needs to have a good memory, propose good possibilities and explore to check the possibility's validity. However, memory and exploration can only provide observation rather than states, as the difference between observation and state in POMDP, which remains challenge still. The agent needs to have reasoning-guided exploration, rather than exhaustive exploration, to quickly find frames that most likely lead to guess, and reason about the underlying hidden states based on the evidence gathered, as explained in the cut-tree example above and the examples in the PDF. Besides, experimentally, we note that with the ideal explorer, the reasoning is still not perfect.
>
> > "embodied vs open-world?"
>
> The difference is subtle. However, in our work, we'd like to emphasize the diverse and rich environment the agent could interact with, while simplifying actuator control, thus the wording.
>
> > "l146: what does a trace look like? l151: "32 distinctive traces"
>
> See pdf attached and supp. "32 distinctive traces" means that there are a total of 32 distinctive kinds of traces that can be left in the environment (see Figure A1).
>
> > "60 abductive reasoning tasks"
>
> No. Abductive reasoning tasks refer to tasks that Conan environment can support.
>
> > "l164-168: example of a path + alternative and why...?"
>
> Also see the attached pdf.
>
> > "why detective same actions"
>
> In the ideal case, the detective only needs to navigate. However, in some cases, the detective needs additional actions, such as meeting blocked roads with stones and needing to perform actions to make tools to break stones. Surely masking other actions will make it easier.
>
> > "l226-227: masks are first employed to encode questions...How does the explorer condition on the question?"
>
> Mask is used to guide the model's attention to the relevant parts of the playground observation. In questions like "Why does the vandal cut the tree", we need a pointer to what "tree" actually refers to. So we use a mask the same size of the map to highlight which tree the question refers to, and extract both the linguistic features and the visual features from the masked map as question encoding.
>
> > "step limit on the explorer?"
>
> Yes. 500. Discussed in line 326-327.
>
> > "l174-177: My understanding is that the agent is the detective?"
>
> Yes.
>
> > "l255: the representation of symbolic map?"
>
> It is a 2D array with dimensions [64, 64], where each number in the array represents the type of object in the position.
>
> > "l307: the ideal explorer get the frames of the ground truth vandal trajectory, and extract every 30 frames? Performance gap due to the keyframe extractor dropping vs. the inaccuracy of the reasoning model?"
>
> It's that the final downsampled length is 30 frames. That corresponds to every 6-9 frame extraction. The context window size is 9x9, which means that no information is lost in this process.
>
> > "sec 4.4: access to the vandal policy pi?"
>
> We did assume the detective shares the same policy with the vandal. It's not so realistic; however, it is also reasonable to assume to some extent that we share the same world model: we all know how to make swords or break the stones. But we admit that it's better to train the detective agent, say in traditional RL methods in the forward way to distill the world knowledge. The work explores this direction with a simple assumption and certainly has room for improvement.
>
> > "l345: vanilla-trans susceptible to insufficient evidence ... it's not pretrained to do question answering?"
>
> A very reasonable explanation. Its failure can be attributed to its weakness in reasoning based on incomplete observations. Will pretraining on other vqa tasks help solve Conan? We are willing to explore this possibility in the future work.
>
> > "AfD doesn't improve on standard"
>
> While the final performance is only silightly better. But compared with other methods, AfD is not directly supervised with relationship from traces and questions and answers like others. Instead, AfD is trained with P(S|O) model (state prediction from observation) and a policy model as in vandal. The final results are obtained from an explicit inference process. Considering this fact, AfD and other directly supervised models reaching similar performance and even slightly better should be considered significant results.
>
> [1] Sherlock
> [2] The scientist as child
> [3] Detecting blickets
> [4] Causal learning mechanisms in very young children

---

> > ### Author Response · Authors · 2023-08-17
> >
> > Dear Reviewer yGBc,
> > We wanted to confirm whether the concerns and questions you raised in your initial review have been successfully addressed in our responses and revisions. We truly value your insights and feedback. Please feel free to post additional questions and comments about our work during the author-reviewer discussion period.

---

### Official Review · Reviewer_2rwT · 2023-07-06

**Soundness:** 3 good
**Presentation:** 4 excellent
**Contribution:** 2 fair
**Rating:** 5
**Confidence:** 4

**Summary:**

The paper introduces Conan, an interactive open-world environment for evaluating the active reasoning abilities of agents. In Conan, agents need to answer questions by actively seeking for evidence and acquiring new knowledge in a setting of incomplete information.
Conan is formulated as a detective game. First, during the initialization of the game, a vandal agent completes a task, leaving traces in the environment. Next, questions are generated for a detective agent to answer. The detective has to answer these questions by actively interacting with the environment to reconstruct the actions of the vandal.

**Strengths:**

* The environment differs from existing benchmarks for its active reasoning and interactive multi-round setting
* The paper is very well written and easy to follow
* Conan is an interesting environment that could spur further research on active reasoning in interactive contexts

**Weaknesses:**

* It is not clear how challenging the task is and how good the performance of the models reported in the paper is. A human study would help the reader understand this critical point.
* A qualitative analysis of the games and the behavior of the different agents would also be needed in the main paper to understand when the agents make mistakes and to what extent the tasks are solvable
* As mentioned in the paper, it looks like the explorer does not supply particularly relevant information to the reasoning model, substantiating the intuition that the bottleneck is there, rather than in the reasoning abilities of the model

**Questions:**

Is a human study or an error analysis feasible?

**Limitations:**

The authors described the limitations of the paper

---

> ### Author Rebuttal · Authors · 2023-08-09
>
> Dear reviewer:
>
> Thank you very much for your thoughtful and detailed review. We are pleased that you found this work interesting and unique from existing benchmarks.
>
> > "It is not clear how challenging the task is and how good the performance of the models reported in the paper is. A human study would help the reader understand this critical point."
> > "Is a human study or an error analysis feasible?"
>
> We have now uploaded a PDF file detailing the problem solving process in case you miss the challenges presented in the work. We kindly refer you to it for detailed explanation. Indeed, conducting a human study would be valuable. However, playing this particular game requires strong prior knowledge and familiarity with the game mechanics. Training is not easy, and it's hard to distinguish what leads to humans' failure in this task. As a result, we are still conducting the human study and would update the results in the discussion phase when we have preliminary results.
> Below we present the error analysis for the "goal" split. We examine the accuracy of Frozenbilm-deberta on various tasks, comparing two groups: reasoning based on the ideal explorer and the TRPO explorer.
> As mentioned in the paper, the ideal explorer is relatively oracle-like, as it not only gathers all the traces left by the vandal but also captures temporal information. Results show that in most cases the model answers more questions correctly on Ideal explorer than on TRPO explorer. Reasoning on ideal explorer fails when the questions cannot be directly "seen" from the environment and not found afterwards. For example, detectives can see a table but can not see what was made. On the other hand, reasoning on TRPO explorer struggles with long-term traces, indicating the fact that it can not cover all traces during exploration. Results from the reward also validate this (see response to Reviewer nDEB). We are going to add more error analysis into the supp. Thanks again for your suggestion!
>
> | **Tasks**   | get_drink       | defeat_cow  | get_apple         | make_stone_pickaxe | place_bed   | place_furnace   | defeat_zombie     | make_stone_sword |
> | ----------- | --------------- | ---------------- | ----------------- | ------------------ | ----------- | --------------- | ----------------- | ---------------- |
> | **Trpo**    | 47.06           | 43.90            | 35.7              | 48.48              | 43.90       | 44.44           | 52.50             | 37.50            |
> | **Ideal**   | 100.00          | 85.37            | 78.57             | 39.39              | 87.80       | 83.95           | 75.00             | 4.17             |
> | **get_lava**    | **defeat_skeleton** | **make_iron_sword**  | **get_coal**          | **get_beef**           | **get_diamond** | **get_stone**       | **place_table**       | **get_wood**         |
> | 50.00       | 46.59           | 56.25            | 50.00             | 53.85              | 40.62       | 36.84           | 39.36             | 36.00            |
> | 66.67       | 82.95           | 28.12            | 45.45             | 42.31              | 84.38       | 47.37           | 91.49             | 96.00            |
> | **make_bucket** | **get_iron**        | **get_water**        | **make_iron_pickaxe** | **make_bed**           | **make_steak**  | **make_wood_sword** | **make_wood_pickaxe** |                  |
> | 35.29       | 28.57           | 45.95            | 56.52             | 47.83              | 46.15       | 50.00           | 40.00             |                  |
> | 73.53       | 46.43           | 54.05            | 52.17             | 39.13              | 50.00       | 64.29           | 55.00             |                  |
>
> > "A qualitative analysis of the games and the behavior of the different agents would also be needed in the main paper to understand when the agents make mistakes and to what extent the tasks are solvable.
> > "As mentioned in the paper, it looks like the explorer does not supply particularly relevant information to the reasoning model, substantiating the intuition that the bottleneck is there, rather than in the reasoning abilities of the model."
>
> Thanks for your suggestion. We do consider adding more analysis and discussion about different agents and their results. We have included some qualitative analysis in Sec 5.2. Besides, some extra experiments has been conducted, see response to Reviewer nDEB for more information.
> To make it short, in the case of empty visual inputs, the reasoning performance is quite low, indicating that the agent is severely limited by the lack of visual information. The TRPO explorer shows a noticeable improvement compared to empty visual inputs, indicating that some exploration is better than none. However, the performance is still relatively low. The ideal explorer achieves significantly better performance, indicating that its ability to gather perfect trace evidence greatly benefits the downstream reasoning task. This highlights the importance of effective exploration.
> However, this is not to say the reasoning part can be considered solved. As can be seen in Table 3, even with an ideal explorer, the reasoner still could not well answer the questions. Also see response to Reviewer nDEB for a detailed discussion.

---

> > ### Comment · Reviewer_2rwT · 2023-08-15
> > **Thanks**
> >
> > I appreciate the response of the author to address my comment. I increased my score by 1 point.

---

> > > ### Author Response · Authors · 2023-08-16
> > > **Response to Reviewer 2rwT**
> > >
> > > We greatly appreciate your follow-up. We would also like to share some preliminary results from our human study. We recruited 10 participants from our subject pool and trained them within the proposed Conan environment. For the "goal" split, they averaged 110.0 exploration steps per question and answered 90.9% of the questions correctly. These findings indicate that humans exhibit more efficient exploration and better reasoning capabilities. We will include further details about this aspect in the revised version.

---

### Official Review · Reviewer_nDEB · 2023-07-07

**Soundness:** 3 good
**Presentation:** 4 excellent
**Contribution:** 4 excellent
**Rating:** 7
**Confidence:** 4

**Summary:**

This paper proposes a benchmark for "active reasoning" titled Conan, where, instead of passively answering questions from provided context, an agent must interactively explore its environment to discover information. The authors differentiate such a task from so-called "passive reasoning" tasks such as video-language understanding where the visual input is directly fed into the model. Here, a model itself is responsible for exploring and attaining input that will help a model answer a question.

Conan is implemented in a Minecraft-like gridworld where a "detective" agent must identify activities that a rule-based, goal-oriented "vandal" agent completed, by using traces of the vandals' behavior in the environment. E.g. if a vandal's goal was to make a wooden pickaxe, it may have cut some trees down, and the task for the model will be to answer why the vandal decided to make a wooden pickaxe.

Two alternative methods are adopted to approach this task: one which uses an explorer agent, trained with an exploration reward, to generate evidence that is fed into a standard VLM, and another, "Abduction from Deduction", where one directly learns to predict the goal of the vandal from an inferred state trajectory.

Overall this is an interesting paper, and I think the dataset and benchmark will be valuable for the community. The models tested for this environment are relatively simple, not really making full use of the "joint exploration and reasoning" abilities that Conan purportedly tests, however. Moreover I do have some outstanding questions, and some issues with the experiments (specifically a missing baseline) that prevent me from assigning a higher score. However I'm open to changing my score after the author response and discussion period.

**Strengths:**

- To my knowledge, the "active reasoning" component of Conan is an important area in vision/lang/RL, and is certainly underexplored in current embodied QA settings (although it is probably implicitly present in instruction following benchmarks and work like SayCan).
- The dataset seems high quality and should be a useful contribution for the field.
- Fairly sound experimental evaluation, comparing a variety of RL explorer agents with a variety of vision-language models.

**Weaknesses:**

- **Missing negative control baselines.** The experimental results should compare evidence gathered by the trained explorer agents to a weaker negative baseline, e.g. a random untrained explorer policy or no explorer input at all. This is needed to convincingly show that the explorer agents are actually learning to explore in a way that is more beneficial than chance, at least for the downstream QA task.
    - In general I'm definitely more interested in to what extent performance is bottlenecked by good exploration on this task, rather than fixing the explorer evidence and trying subtly-differently-trained large VLMs.
- As mentioned by the in the conclusion (L376-377), one of the key promises of Conan is that strongest performance on this benchmark should intuitively be achieved by models that jointly learn to reason and explore, but right now a decoupled two-stage process is adopted—first training an exploration model with an oracle "find trace" reward, extracting relevant keyframes, then training a VLM on top of the frozen policy. As a result, models trained for this task (for now) look like typical embodied QA models doing passive reasoning (just with the extra step of training an explorer to get the vision input). Of course, it's not strictly necessary for a benchmark paper to include an strong novel model for the benchmark, but without it we aren't able to evaluate whether Conan performance numbers have **headroom** for more sophisticated approaches to joint exploration and reasoning. (It seems like this should be the case, but we can't really be sure).
- Some more outstanding questions (see next section)


**Questions:**

- The idea of answering questions via "traces" left behind in the environment is a little hard to grasp and could use more explaining. There are some example traces in Figure 1b but these are small figures and aren't explained that much. The authors could spend some more time walking through example "traces" and more fully convince readers that the traces indeed provide answers to some of the more subtle questions in the benchmark (e.g. how do traces indicate that the vandals goal is to "get the diamond", couldn't the vandal have alternative goals like get iron/coal?).
	- It might be valuable to have an "oracle" setting to demonstrate that with perfect trace evidence and enough training models can 100% this task—or else explain why such an oracle doesn't exist (e.g. what causes the lagging performance even for the "Ideal Explorer" setting in Table 3? Is it an amount of training data issue? Are there still limitations as to the quality of the trajectory attained even by the ideal explorer, and is there an even more suitable oracle for this?)
	 - Related to this, can authors further explain L308 ideal explorer: "visible to the ground-truth vandal’s trajectory"? What does "visible to" mean—shouldn't the ideal explorer literally *be* the ground-truth vandal's trajectory?
- More details could be provided on the specific kinds of generalization splits tested here. At test time, is there a subset of abductive reasoning tasks held out that the model has never seen? Or does the model see the same questions it has seen during training, just in unseen environments? Do the test environments differ systematically from those during train (or is it possible to induce such a )
	- Evaluating (and supporting) such compositional/systematic generalization splits would greatly increase the appeal of the benchmark.
 - (Related to the above) Do authors have an intuition as to whether test-time performance drop is more due to the VLM's inability to generalize, or the explorer's inability to generalize and generate good evidence for unseen enviroments (if test does indeed have unseen environments)? Is it possible to disentangle the two using different evaluation splits in the environment?
 - In Figure 4, does a score of near 100 as attained by eg TRPO indicate that the explorer indeeds find all traces (+100) reward all of the time? If not, is there an explicit measure of that?

**Limitations:**

Yes

---

> ### Author Rebuttal · Authors · 2023-08-09
>
> Dear Reviewer,
> Thank you very much for your thoughtful and detailed review.
> > "Missing negative control baselines."
>
> Thank you for your insightful feedback. We have tried our model with empty visual inputs as a negative control baseline as your suggestion and we show the results on "goal" split here:
> || Vanilla-Trans | F-BiLM-BERT | F-BiLM-DeBERTa | Flamingo-mini |
> |-| -|-| -|- |
> | Empty visual inputs | 26.4| 25.5| 25.9| 22.9|
> | Trpo explorer | 25.0 | 44.4| 43.1 | 43.3 |
> | Ideal explorer| 78.4| 59.5| 71.8| 47.8|
>
> The results show that using empty visual inputs yields random performance across all settings. Besides, it also shows that the training QA pairs are unbiased. The TRPO explorer achieves higher performance, which suggests that the exploration strategy learned by TRPO helps gather some informative evidence for the reasoning process. The ideal explorer is an oracle-like exploration policy that has access to perfect trace evidence and temporal information. It provides the most comprehensive information about the environment. This highlights the importance of effective exploration in improving reasoning performance.
> However, it does not mean that reasoning is less important, as even with the ideal explorer, the model still could not achieve satisfactory performance. Based on all results, collecting informative evidence seems to be more important in the overall objective.
>
> > "a decoupled two-stage process is adopted"
> > "more sophisticated approaches to joint exploration and reasoning"
>
> Due to the word limit, we kindly refer you to our response to Reviewer u9zC for a detailed discussion.
>
> > "answering questions via "traces" left is a little hard to grasp"
> > "some example traces in Figure 1b aren't explained that much"
>
> Thanks for your kind suggestion. We kindly refer you to the attached file for a simple demonstration of traces and reasoning steps. We are also considering adding more comprehensive examples in the supp. Thanks again for your suggestion!
>
> > "traces indicate that the vandals goal is to "get the diamond" or other alternative goals"
>
> For instance, if traces indicate that the vandal made a pickaxe, its intent could be to get iron or diamond according to the task dependency graph. As a result, the detective should continue to find more distinctive traces that can help exclude the wrong possibility, say checking the mine area.
>
> > “an "oracle" setting to demonstrate that with perfect trace evidence and enough training models can 100% this task”
> > "limitations of the ideal explorer traces"
> > "L308 visible to the ground-truth vandal’s trajectory”
>
> The ideal explorer provides the vandal's trajectory, while it cannot directly observe the actual vandal states when an action is taken, but only the traces the action left. This can introduce confounding factors, leading to potential confusion, necessitating the reasoning component in connecting the traces and figuring out the states. That's why we design Conan: it is important not only to explore the traces, but also to reason the hidden states. Agents must engage in abductive reasoning based on the traces they observe to reconstruct the entire story, which is indeed a challenging task. It could be a data problem if one takes on a data-driven perspective. The ideal explorer is good enough, so in this case we tend to believe it is the reasoner to blame.
>
> > "Is there a subset of abductive reasoning tasks held out that the model has never seen? Or does the model see the same questions it has seen during training, just in unseen environments? Do the test environments differ systematically from those during train?"
> > "Evaluating (and supporting) such compositional/systematic generalization splits"
>
> The test environments are disjoint from those during training, though they are sampled from the same distribution. But test questions are from the same distribution as training and could be similar. As the results show, even on IID questions, models still fare worse than expected. Therefore, as the first attemtp in active reasoning, we primarily focus on the current setting, and would like to see joint community efforts when it is time to introduce other generalization in the problem.
>
> > "Performance drop due to the VLM's inability to generalize, or the explorer's inability to generalize and generate good evidence for unseen environments? Possible to disentangle them?"
>
> We tend to believe the answer is complicated, but the intuition is that incomplete information plays the more critical role. Evidence above and in the paper suggests that compared to the reasoning component, the exploration component may be more important in order to achieve "OK" performance. But to reach perfection, the juice in the long tail can only be squeezed out by a good reasoner. Our current setting (test environments are unseen during training) shows that with the ideal explorer, performance could be greatly improved, meaning that the VLM could generalize but the explorer couldn't.
>
> > "Does a score of near 100 indicate that the explorer indeeds find all traces? Explicit measure?"
>
> Not exactly. As mentioned in section 4.1, the agent receives a step reward when it discovers more traces, which is intentionally designed to avoid sparse rewards. Additionally, if the agent successfully finds all traces left by the vandal, it will receive another 100 reward. Since there can be different environments and the vandal can leave long traces (note that there are steps with a reward of 2), the total step reward may vary slightly. We check statistics to explain this issue. For the TRPO explorer, it manages to find all traces in only about 1% of the environments. The ideal explorer achieves an average reward of 123.12 in 10000 envs. Treating it as the lower bound, the TRPO explorer is still imperfect. Careful analysis shows that TRPO only manages to find all traces in 1% of the environments.
>
> [1] Kurt Spencer. Noise!, 2014. URL https://github.com/KdotJPG/OpenSimplex2

---

> > ### Comment · Reviewer_nDEB · 2023-08-15
> > **Thanks**
> >
> > Thanks to authors for their detailed rebuttal. I appreciate the follow-up experiments and negative control baselines which verify that the models are indeed improving by using the image input. However I do think the random explorer baseline is also important, as it's possible to get some evidence by just randomly exploring (apologies if I did not sufficiently emphasize this in my initial review).
> >
> > Regardless, I think the other responses to the rebuttal also increase my confidence in the paper—I'll increase my score to a 7.

---

> > > ### Author Response · Authors · 2023-08-16
> > > **Response to Reviewer nDEB**
> > >
> > > Thank you very much for your follow-up! We are pleased to provide some preliminary results while training our models with the random explorer. Also on the "goal" split, we have observed the following accuracy rates: Vanilla-Trans 25.9%, F-BiLM-BERT 35.8%, F-BiLM-DeBERTa 41.6%, and Flamingo-mini 38.0%. These results show that the performance with the random explorer is notably improved compared to scenarios with empty visual inputs. Interestingly, we also observe that while the random explorer greatly falls short of the ideal explorer, its performance is only slightly lower than that of the TRPO explorer. Notably, our agents are initialized at the starting point of the traces, which indicates that they can collect some traces around them through random exploration. This observation suggests that agents do indeed benefit from the presence of traces, although they may be collected randomly. Besides, while the trained TRPO explorer exhibits certain effectiveness, it still has a long way to go.

---

> > > > ### Comment · Reviewer_nDEB · 2023-08-16
> > > > **Thanks**
> > > >
> > > > It's great to see these follow up experiments! It does seem like random exploration is doing well and that there's more headroom to grow from TRPO. These results increase my confidence in the paper and I would encourage their inclusion in the paper.

---

> > > > > ### Author Response · Authors · 2023-08-17
> > > > >
> > > > > Thank you for your follow-up! We will include these results in the paper. We are glad to have more discussions in case you have any further concerns.

---

### Official Review · Reviewer_i6nS · 2023-07-22

**Soundness:** 2 fair
**Presentation:** 2 fair
**Contribution:** 3 good
**Rating:** 5
**Confidence:** 2

**Summary:**

To address the gap in handling incomplete-information questions, this paper introduces an interactive open-world environment called "Conan." The purpose of Conan is to motivate and evaluate agents' active reasoning ability by requiring them to explore, gather evidence, and combine knowledge to solve complex scenarios. The ultimate goal is to improve AI agents' ability to interact with and learn from the world.

**Strengths:**

This paper introduces a new interactive open-world environment called "Conan." The purpose of Conan is to motivate and evaluate agents' active reasoning ability by requiring them to explore, gather evidence, and combine knowledge to solve complex scenarios. The ultimate goal is to improve AI agents' ability to interact with and learn from the world.

**Weaknesses:**

The author claims that the detective spawned in the environment needs to answer all questions through exploration, connecting key frames, and reaching conclusions. Yet, some questions presented in the Figures/Tables can be answered without exploring the environment. The process of answering questions through exploration requires more clarity and elaboration.

Regarding the reward function used in the task, the author mentions that it incentivizes the agent to search for clues and traces related to the given question. However, the paper lacks detailed information about how the reward is designed and the specific motivation behind it.

The writing appears to be cumbersome and illogical, making it difficult to grasp the actual meaning of the task. The purpose of introducing this new task is not clearly stated, leaving readers with questions about its significance and potential applications.

In conclusion, while the paper introduces an intriguing concept of an interactive open-world environment for AI agents, there are aspects that require further clarification and refinement. Addressing the limitations and providing more detailed explanations about the task and reward design would enhance the overall understanding and impact of this research.

**Questions:**

See Weaknesses

**Limitations:**

See Weaknesses

---

> ### Author Rebuttal · Authors · 2023-08-09
>
> Dear Reviewer:
>
> Thank you very much for your review and positive rating!
> > "Some questions presented in the Figures/Tables can be answered without exploring the environment. "
>
> No. All of the questions have multiple possible answers and need to be inferred from traces in the environment. From the detective's view, he cannot directly "see" what happened in the environment. For example, when asked "Why did the vandal cut a tree? ", the detective needs to find traces before and after (especially after) to see whether the vandal needs some woods to make tools or just want to collect apples to eat. If the reviewer believes some questions could be answered without exploration, please point it out in the discussion phase and we are happy to help further address the concern.
>
> > "However, the paper lacks detailed information about how the reward is designed and the specific motivation behind it."
>
> We apologize for the ambiguity. As mentioned in Sec 4.1,  the agent will receives a positive reward of 1 when a trace first appears within its local view or 2 when the trace is closely associated with the question. The agent receives a reward of 100 if it successfully finds all traces left by the vandal. Additionally, the agent incurs a penalty of -0.1 for every timestep it takes. By rewarding the agent for successfully identifying and following the traces, we foster behavior that mimics real-world detective work, where finding clues is a fundamental aspect of solving mysteries.
>
> > "The writing appears to be cumbersome and illogical, making it difficult to grasp the actual meaning of the task. The purpose of introducing this new task is not clearly stated, leaving readers with questions about its significance and potential applications."
>
> Thank you for your feedback on the writing. We apologize for any confusion. We will make the necessary improvements to address these concerns and provide a more concise and logical presentation. In a nutshell, the task is designed for capturing humans' active and exploratory nature in abductive reasoning in the real world and providing agents with an open-world environment which encourages active exploration and multi-round abductive inference based on in-situ gathered evidence and existing knowledge. The task goes beyond existing single-round passive tasks and contributes to the broader pursuit of building more intelligent and human-like AI systems.
>
> > "providing more detailed explanations about the task and reward design"
>
> Thanks for your kind suggestion. We kindly refer you to the attached file for a simple demonstration of traces and reasoning steps. We are also considering adding more comprehensive examples in the supp. Thanks again for your suggestion!

---

### Official Review · Reviewer_u9zC · 2023-07-26

**Soundness:** 3 good
**Presentation:** 4 excellent
**Contribution:** 3 good
**Rating:** 6
**Confidence:** 5

**Summary:**

This paper introduces Conan, an interactive environment as a benchmark to evaluate agent’s active abductive reasoning abilities to answer questions in an incomplete (or partial) information scenario. Because of partial information, the model requires further exploration in the scene to answer the questions which is posed as a detective game. The paper also proposes the Abduction from Deduction (AfD) approach which relies on Bayesian statistics. The experimental evaluation showcases the strengths and weaknesses of various machine learning models in different settings.

**Strengths:**

- The paper is well-motivated and clearly written in most parts.
- The authors promise the availability of corresponding code and a detailed description of hyperparameters which would be essential for reproducibility if open-sourced.
- The paper introduces the AfD approach using Bayesian rules which is interesting to the research community.
- Relevant baselines have been selected to comprehensively evaluate various RL and multi-modal models.


**Weaknesses:**

- The proposed environment looks like a toy benchmark with limited real-life applications.
- Dataset statistics is missing in the work which might expose data biases related to goal, intents or survival questions.
- The question templates are also limited and it is not clear if the model learns to take advantage of any data biases to answer the questions. It is also not clear if the questions are asked in a particular order.
- This is framed as a multi-choice QA task with limited generalizability.
- The proposed approach lacks a close integration between exploration and reasoning processes where the training and evaluation for different models is independent.


**Questions:**

- Can you elaborate on how this research can be improved to integrate exploration and reasoning processes in future work?
- What was the main motivation behind maintaining the detective set invincible, ie. not maintaining the survival status on line 178?
- On line 363, the authors mention that the proposed approach is akin to “human-like reasoning”. It is not clear as to how solving this toy task with an independent explorer and reasoning model brings us closer to “human intelligence”

Suggestions:
- It would help to make the best results in bold in Table 3.







**Limitations:**

See weaknesses for more information about the limitations

---

> ### Author Rebuttal · Authors · 2023-08-09
>
> Dear Reviewer:
>
> Thank you very much for your thoughtful review! We would like to discuss your concerns here:
> > "look like a toy benchmark with limited real-life applications"
>
> It is true that Conan is synthetic and may appear simplistic at first glance. However, the primary goal of Conan is not to present real-world complexity but rather to introduce the active reasoning setting to the community. Direct applications may not be straightforward; however, we believe the autonomocy in a successful Conan solver would be utterly desirable in our never-ending pursuit of AI. For example, we have many real world scenearios where we need informed search to draw conclusion, say new material discovery (like LK-99), causual discovery, or new drug synthesis. Though Conan is more about common sense, the general problem should be very similar. Besides, "toy" as it may seem, Conan, as Reviewer yGBc puts it, is already quite significant and non-trivial for AI systems to master. From our perspective, the environment already covers all the critical aspects in active reasoning. The toy-ish nature simplifies researchers' efforts to control and explore various aspects of the problem when attempting to address the difficulties without concerning about the real-world complexities, as the research history on CLEVR[1], CLEVRER[2] and EQA[3] shows (despite them being simpler than Conan). Successfully solving these tasks should be the grounds for developing more general and adaptable AI models in the future when we envision autonomous machine and human-machine collaboration. We believe that treating Conan as "toy" might overlook the true challenges and benefits it offers.
>
> > "Dataset statistics is missing"
>
> Thanks for the reminder. Please see dataset statistics below:
> |Category|Train|Test|Val|A|B|C|D|
> | -|-|-|-|-|-|-|-|
> |Intent|71162|9152|8822|24.99%|25.20%|24.89%|24.93%|
> |Goal|8000|1000|1000|24.89%|25.08%|24.87%|25.16% |
> |Suvival|7365|1560|1596|25.13%|24.95%|24.95%|24.97% |
>
> |Tasks|get_drink|defeat_cow|get_apple|make_stone_pickaxe|place_bed| place_furnace|defeat_zombie|make_stone_sword |
> |-|-|-|-|-|-|-|-|-|
> |Percentage|2.47|8.49|2.52|2.87|8.44 | 8.23 |7.8|2.65|
> |**get_lava**|**defeat_skeleton**|**make_iron_sword**|**get_coal**|**get_beef**|**get_diamond**|**get_stone**|**place_table**|**get_wood**|
> |2.72 |8.7|2.64|2.42|2.7|2.39|2.67|8.24|2.67|
> |**make_bucket**|**get_iron**|**get_water**|**make_iron_pickaxe**|**make_bed**|**make_steak**|**make_wood_sword**|**make_wood_pickaxe**||
> |3.11|2.44|2.2|2.95|2.71|2.81|2.53|2.63||
>
> We will release more detailed dataset statistics together with the benchmark.
>
> > "The question templates are also limited"
> > "if take advantage of any data biases / questions are asked in a particular order"
>
> Almost all VQA tasks do suffer from this issue as question templates are always limited. Our approach involved carefully crafting and curating a diverse set of question templates that cover various aspects of the task. Questions are not asked in a particular order. The choices are also randomly sampled from a pool and shuffled. Experiments have shown that without any visual input, the models only achieved random-level performance (see response to Reviewer nDEB), showing that templates do not introduce biases.
>
> > "framed as a multi-choice QA task with limited generalizability."
>
> Conan can be designed as an open QA task. However, we design Conan as a multi-choice QA task for easier and clearer evaluation.
>
> > "the main motivation behind maintaining the detective set invincible"
>
> The decision was driven by the design goal of focusing on active exploration and reasoning rather than adding to the challenge by survival. In this way, we can ensure that the main emphasis remains on solving complex reasoning tasks and answering questions related to the visual scenes provided. Besides, the task will be too hard if the detective may be dead (but it can be if you change the hyperparam).
>
> > "integrate exploration and reasoning processes"
> > "how an independent explorer and reasoning model brings us closer to “human intelligence”
>
> We acknowledge this limitation and appreciate your interest for improvement. Ideally, we should jointly optimize both exploration and reasoning components. As you said, it's more human-like and intelligent, for instance, the reasoner can provide feedback to the explorer, guiding it towards more informative exploratory actions that are likely to yield relevant traces.
> However, there are significant obstacles in practice (we tried but failed). Attributing credit to exploratory decisions that have long-term consequences can be complex. When exploration actions yield results much later, it becomes difficult for the model to understand the causal relationship between the exploratory decisions and their eventual impact on reasoning and answering questions. This highlights the interdependence between the exploration and reasoning processes. Improving one aspect requires advancements in the other, creating mutual dependency that complicates the optimization process. The reasoning component itself requires significant training and computational resources, especially when based on large language models. The need for substantial computational power makes the joint optimization of exploration and reasoning even more challenging.
> From a community perspective, it's also a common practice [3-5]. So we adopt this route. We believe future work on Conan shall prioritize reasoning over exploration: not to say that exploration is less important, but guided by a reasoning engine. Similar to some forms of (soft) tree search, we hope that a reasoner would propose possible subgoal and the explorer would explore the environment conditioned on the subgoal, and the entire process may be integrated in a MCTS-based learning framework like AlphaGo.
>
> [1] Clevr
> [2] Clevrer
> [3] Embodied question answering
> [4] Embodied question answering in photorealistic environments with point cloud perception
> [5] Iqa

---

> > ### Comment · Reviewer_u9zC · 2023-08-17
> >
> > I have read the author's response and fellow reviewer's feedback. I believe that the authors have addressed most of my concerns. Particularly, I also liked the point raised by Reviewer nDEB and the author’s experiments related to the random baselines.

---

> > > ### Author Response · Authors · 2023-08-17
> > > **Thanks**
> > >
> > > Thank you for your follow-up. We are glad to have more discussions if you have any further concerns.

---

### Author Rebuttal · Authors · 2023-08-10

To all reviewers:
We are sincerely appreciative and grateful for the time each of you have spent reading our work and giving useful, thoughtful and constructive feedback. The feedback is substantial and quite helpful for improving our paper. In particular, we would like to thank reviewers for acknowledging our work to be well-motivated (Reviewer u9zC, i6nS, nDEB, yGBc), high quality and should be a useful contribution (Reviewer nDEB), an interesting environment that could spur further research on active reasoning in interactive contexts (Reviewer 2rwT) and an ambitious benchmark trying to tackle an exciting and interesting problem (Reviewer yGBc). We are very glad to see that there is a general interest in such a topic and do hope that this work serves as a valuable contribution to the community.

---

### Comment · Area_Chair_QC9B · 2023-08-20
**Thanks for your response -- AC Comment**

Thanks for providing your responses to the reviewers' comments. They are comprehensive and answer many of the concerns raised during the initial review phase. The discussion with other reviewers also provides additional context for the questions raised during the initial review.

-- Your AC

---

> ### Author Response · Authors · 2023-08-20
>
> We would like to express our gratitude to the AC and all reviewers for their efforts in evaluating our work. Your feedback and suggestions have played a pivotal role in improving our work. We are glad to address any future concerns or questions that may arise.

---

### Decision · Program_Chairs · 2023-09-21

**Decision:**

Accept (poster)

**Comment:**

This work received primarily positive reviews from the reviewers. Several questions were raised during the review period, including about baselines, evaluation setup, and claims made in the paper. The authors provided a rebuttal that addressed many of these concerns effectively. The rebuttal contained several key pieces of information, including additional experiments, which provide additional quantitative support for the benchmark and a better understanding of what capabilities might be required to solve it. The authors are strongly encouraged to incorporate the information from the rebuttal into the final version for completeness.